# Entry of spores into intestinal epithelial cells contributes to recurrence of *Clostridioides difficile* infection

Pablo Castro-Córdova [1,2], Paola Mora-Uribe[1], Rodrigo Reyes-Ramírez[1,2], Glenda Cofré-Araneda[1], Josué Orozco-Aguilar[1,2], Christian Brito-Silva[1,2], María José Mendoza-León [1,2], Sarah A. Kuehne[3], Nigel P. Minton [4], Marjorie Pizarro-Guajardo[1,2,5] & Daniel Paredes-Sabja [1,2,5✉]

*Clostridioides difficile* spores produced during infection are important for the recurrence of the disease. Here, we show that *C. difficile* spores gain entry into the intestinal mucosa via pathways dependent on host fibronectin-$\alpha_5\beta_1$ and vitronectin-$\alpha_v\beta_1$. The exosporium protein BclA3, on the spore surface, is required for both entry pathways. Deletion of the *bclA3* gene in *C. difficile*, or pharmacological inhibition of endocytosis using nystatin, leads to reduced entry into the intestinal mucosa and reduced recurrence of the disease in a mouse model. Our findings indicate that *C. difficile* spore entry into the intestinal barrier can contribute to spore persistence and infection recurrence, and suggest potential avenues for new therapies.

[1] Microbiota-Host Interactions and Clostridia Research Group, Facultad de Ciencias de la Vida, Universidad Andrés Bello, Santiago, Chile. [2] ANID – Millennium Science Initiative Program - Millennium Nucleus in the Biology of the Intestinal Microbiota, Santiago, Chile. [3] School of Dentistry and Institute for Microbiology and Infection, University of Birmingham, Birmingham, UK. [4] BBSRC/EPSRC Synthetic Biology Research Centre, School of Life Sciences, Centre for Biomolecular Sciences, The University of Nottingham, Nottingham, UK. [5] Department of Biology, Texas A&M University, College Station, TX, USA. ✉email: dparedes-sabja@bio.tamu.edu

C lostridioides difficile is a strict anaerobic Gram-positive pathogenic bacterium that forms highly resistant spores that easily persist in the environment and contribute to the transmission of C. difficile infections (CDI) through fecal–oral route[1]. Disruption of the gut microbiota by broad-spectrum antibiotics leads to an optimal environment for C. difficile colonization and proliferation in the colon and disease manifestation. CDI currently leads hospital-acquired diarrhea associated to antibiotics in the United States and worldwide[2]. In the US alone, ~500,000 patients per year become infected with CDI, and mortality rates reach ~8% of total patients[2]. The annual cost of CDI to the health care system is estimated at ~US 4.8 billion[2]. Treatment of CDI usually involves antibiotic therapy, typically vancomycin or metronidazole and, most recently, fidaxomicin[2], which, although resolves the infection in ~95% of the cases, leads to recurrence of CDI (R-CDI) in 15–30% of the individuals[3–5].

During infection, C. difficile produces two major virulence factors, toxins TcdA and TcdB, responsible for the clinical manifestation of the disease, induce pro-inflammatory cytokines, disruption of tight junctions, detachment of intestinal epithelial cells (IECs), and loss of transepithelial barrier[6]. C. difficile also initiates a sporulation pathway that leads to the production of new metabolically dormant spores in the host's intestine[1]. In vivo, spore formation is essential for the recurrence of the disease[7]. Moreover, spore-based therapies that remove C. difficile spores from the intestinal mucosa contribute to reducing the recurrence of the disease in animal models[8].

Recent in vivo studies in the laboratory strain 630 suggest that the spore-surface mucus-binding protein, peroxiredoxin–chitinase CotE, and the exosporium collagen-like BclA1 proteins are required for the colonization and infectivity in a mouse model of CDI[9,10]. However, the surface layer of 630 spores does not resemble that of clinically relevant strains, which exhibit hair-like projections in their spore surface, structures that are absent in strain 630[1,11,12]. Notably, most clinically relevant sequenced C. difficile isolates, including isolates of the epidemically relevant 027 ribotype, have a truncated bclA1 due to a premature stop codon in the N-terminal domain[13], resulting in the translation of a small polypeptide, which localizes to the spore surface[10]; thus limiting the breadth and depth of these results.

C. difficile spores exhibit high levels of adherence to IECs in vitro[14,15], and that the hair-like projections of C. difficile spores come in close proximity with the microvilli of differentiated Caco-2 cells; furthermore, C. difficile spores interact in a dose-dependent manner with fibronectin (Fn) and vitronectin (Vn)[15], two extracellular matrix proteins used by several enteric pathogens to infect the host[16,17]. However, the mechanisms that underline how these interactions contribute to C. difficile spore persistence in vivo and contribute to the recurrence of the disease remain unclear.

Herein, we first demonstrate that C. difficile spores gain entry into the intestinal epithelial barrier of mice and that spore entry into IECs requires serum molecules, specifically Fn and Vn, that are luminally accessible in the colonic mucosa. We also demonstrate that the spore entry pathway is Fn-$\alpha_5\beta_1$ and Vn-$\alpha_v\beta_1$ integrin-dependent. Next, we show that the spore-surface collagen-like BclA3 protein is essential for spore entry into IECs through these pathways in vitro and for spore adherence to the intestinal mucosa. Our results also show that BclA3 contributes to the recurrence of the disease in mice. We also observed the therapeutic potential of blocking spore entry into the intestinal epithelial barrier, and how coadministration of nystatin with vancomycin reduces spore persistence and R-CDI in mice. Together, our results reveal a novel mechanism employed by C. difficile spores that contributes to R-CDI, which involves gaining intracellular access into the intestinal barrier via BclA3-Fn-$\alpha_5\beta_1$ and BclA3-Vn-$\alpha_v\beta_1$ specific, and that blocking spore entry contributes to reduced recurrence of the disease.

## Results

### C. difficile spores internalize into the intestinal barrier in vivo.
To study the interaction of C. difficile spores and the host's intestinal barrier, we used a colonic/ileal loop assay infected with C. difficile spores for 5 h[18], where C. difficile R20291 spores were labeled with anti-spore antibodies[13,18]. We observed similar levels of adherence of C. difficile spores to the colonic and ileum mucosa (Fig. 1a–c), with no preference for the site of spore adherence in both colonic and ileum mucosa (Fig. 1d, and Supplementary Figs. 1 and 2). Strikingly, we observed that C. difficile spores were able to cross the mucosal barrier in the colonic/ileal loop assay (Fig. 1a, b, e, f, Supplementary Movies 1 and 2, and Supplementary Figs. 3 and 4). We observed that 4.6 and 3.7 spores per $10^5 \, \mu m^2$ were able to cross the mucosal barrier in colonic and ileal loops (Fig. 1e), corresponding to 0.92 ± 0.30% and 1.04 ± 0.48% of the total spores, respectively. In the colonic mucosa, internalized C. difficile spores were found to homogeneously localize 10–30 μm from the colonic surface and 5–50 μm from the closest crypt membrane, while in the ileum mucosa spores were homogeneously found at 15–70 μm from the villus tip and 10–50 μm from the villus membrane in ileal loops, (Supplementary Fig. 5), indicating multiple sites of entry in colon and ileum.

### C. difficile spore entry into intestinal epithelial cells requires serum components in vitro.
Our previous in vitro studies in IECs were conducted in the absence of fetal bovine serum (FBS) and did not evidence internalized spores[14,15,18,19]. Therefore, we assess if FBS contributed to spore entry by confocal fluorescence microscopy by analyzing monolayers of polarized T84 IECs (Fig. 2a and Supplementary Fig. 6a, b) and differentiated Caco-2 cells (Supplementary Fig. 6c, d), which were infected with C. difficile spores of the epidemically relevant R20291 and the commonly used strain 630 in the presence of FBS. In both cell lines, several intracellular spores of strain C. difficile 630 were found to be located between the apical and basal actin cytoskeleton (Fig. 2a and Supplementary Fig. 6). To obtain convincing evidence of entry of C. difficile spores into IECs, we analyzed polarized monolayers of T84 and Caco-2 cell lines infected with C. difficile 630 or R20291 spores using transmission electron microscopy (TEM). Electron micrographs evidence that some C. difficile spores were found extracellularly in the apical membrane, while others were found intracellularly (Fig. 2b–d). Intracellular C. difficile spores were surrounded by an endosomal-like membrane (Fig. 2c, d). Notably, the formation of membrane lamellipodia-like protrusions and circular ruffle surrounding C. difficile 630 spores were evidenced at the site of attachment of C. difficile spores to the apical membrane (Fig. 2e), suggesting macropinocytosis-like endocytosis of C. difficile spores. Intracellular spores of the strain, R20291, were also evidenced in differentiated Caco-2 cells (Fig. 2f–h).

Next, to quantitatively assess the internalization of C. difficile spores into non-phagocytic cells, we developed an exclusion assay in which, in nonpermeabilized cells, only extracellular spores are fluorescently labeled with anti-C. difficile spore antibody, while total spores can be quantified by phase-contrast microscopy; intracellular spores are not stained by anti-C. difficile spore antibody (absence of fluorescence) and are only detectable by phase-contrast microscopy (Supplementary Fig. 7a). With this assay, we probed that entry of 630, and R20291 spores into monolayers of Caco-2, T84, Vero, and HT29 cell lines significantly increased in the presence of FBS (Fig. 2i, j), as well as with serum from various mammalian

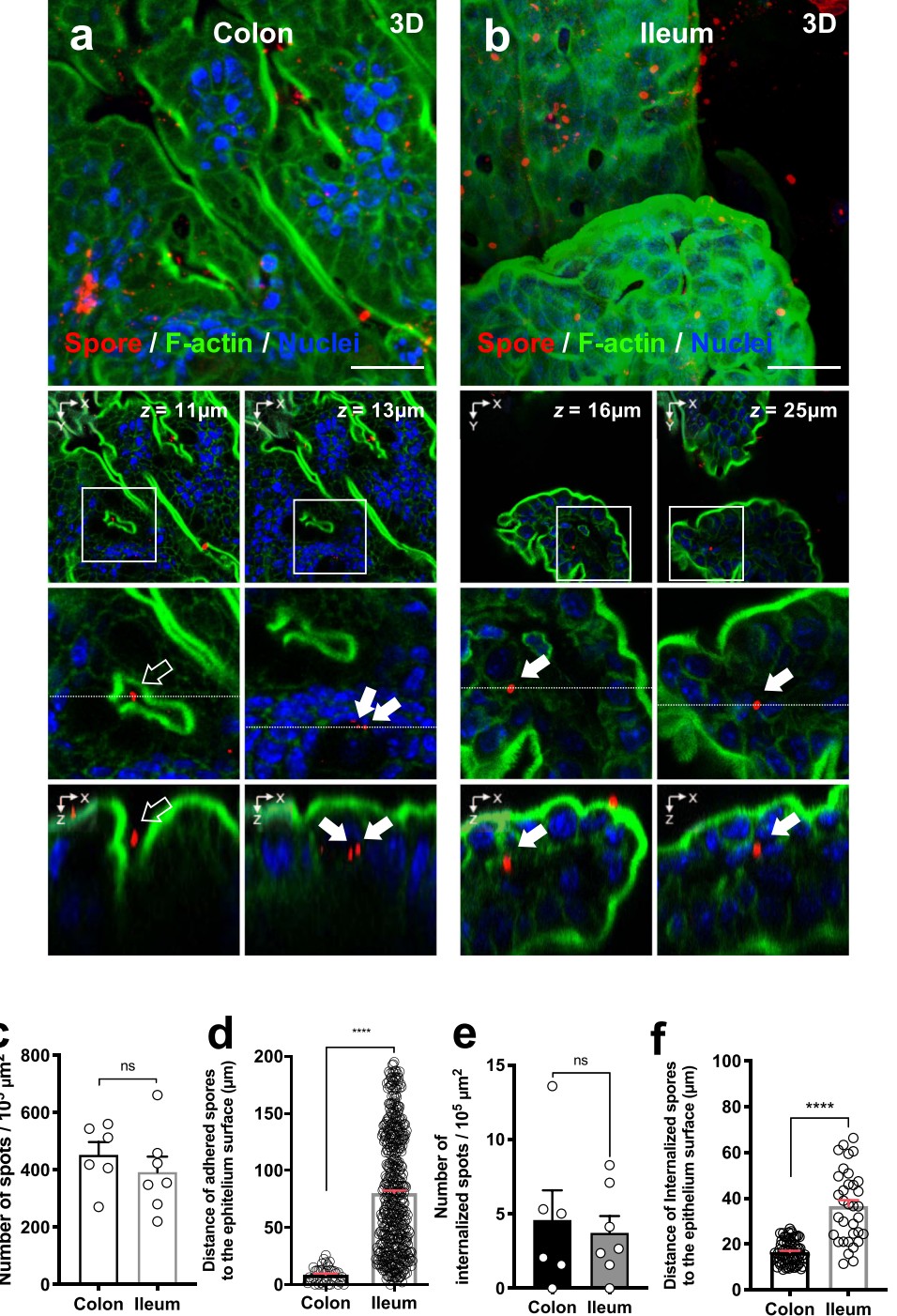

**Fig. 1 C. difficile spore adherence and internalization into intestinal barrier in vivo. a, b** Representative confocal micrographs of fixed whole-mount of **a** colonic, and **b** ileum mucosa of C57BL/6 mice infected with $5 \times 10^8$ C. difficile R20291 spores for 5 h. C. difficile spores are shown in red, F-actin is shown in green and nuclei in blue (fluorophores colors were digitally reassigned for a better representation). Micrographs are representative of $n = 3$ independent mice. **c** Adherence of C. difficile spores to colonic and ileum mice mucosa obtained of four fields from $n = 6$ and $n = 7$ tissues of independent mice, respectively. **d** Distance of $n = 499$ and $n = 40$ adhered spores from the colonic epithelial apical surface or from the villus tip respectively visualized of two fields from $n = 2$ mice each. **e** Quantification of internalized C. difficile spores in the colonic and ileum mucosa obtained of mice described in **c**. **f** Distance of internalized spores from the epithelium surface for the colonic mucosa ($n = 71$) or from the villus tip to the ileum mucosa ($n = 34$) obtained of mice described in **c**. White arrow and empty arrow denote internalized and adhered C. difficile spores, respectively. Scale bar, 20 µm. Error bars indicate the mean ± S.E.M. Statistical analysis was performed by two-tailed Mann–Whitney test; ns, $p > 0.05$; ****$p < 0.0001$.

species (Fig. 2k). The percentage of internalized spores of 630 and R20291 strains was highest at 5 h post infection in Caco-2 and T84 cells (Supplementary Fig. 7b, c). Spores of various clinically relevant ribotypes were able to internalize into Caco-2

cells (Supplementary Fig. 7d). Overall, these results demonstrate that C. difficile spores are able to gain intracellular entry into non-phagocytic cells and that spore entry is serum-dependent in vitro.

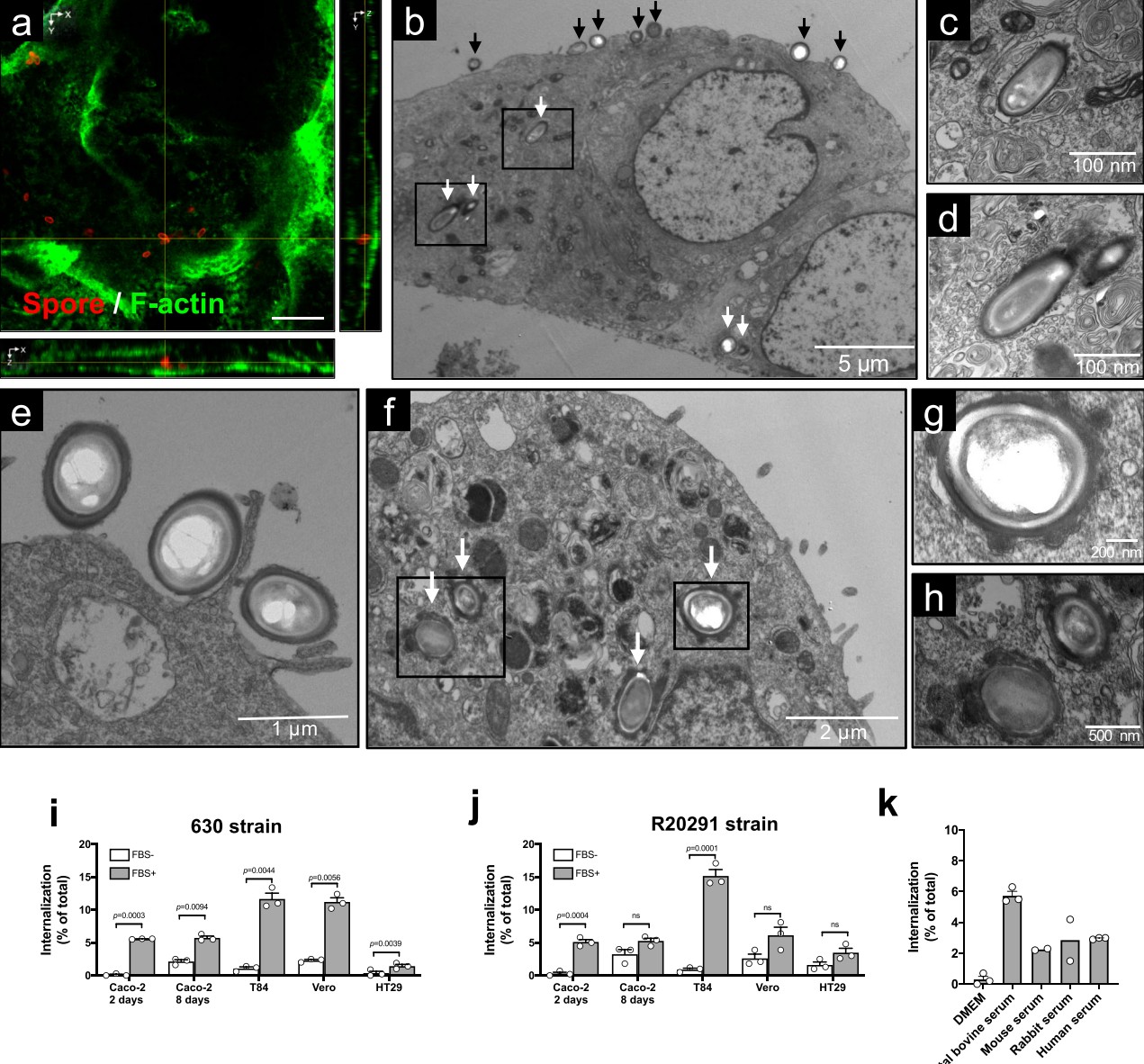

**Fig. 2 *C. difficile* spores are internalized by intestinal epithelial cells. a** Representative confocal micrograph of an internalized *C. difficile* 630 spores in T84 cells. *C. difficile* spores are shown in red, F-actin is shown in green (fluorophores colors were digitally reassigned for a better representation). The images were acquired from $n = 8$ fields from two independent experiments. Yellow lines indicate an internalized spore. **b–e** TEM of differentiated monolayers of T84 cells infected with *C. difficile* 630 spores. Black and white arrows denote extracellular and intracellular *C. difficile* spores, respectively. **c, d** Magnifications of black squares of **b**. **e** An adhered *C. difficile* spore and an apical membrane extension of T84 cells surrounding *C. difficile* spores. **f–h** TEM of differentiated monolayers of Caco-2 cells infected with *C. difficile* R20291 spores. White arrows in **f** indicate internalized *C. difficile* spores. **g, h** Magnifications of black boxes in **f**. Micrographs are representative of $n = 3$ independent experiments. Internalization of *C. difficile* spores **i** strain 630 and, **j** R20291 preincubated with FBS or culture media in undifferentiated (2 days) and differentiated (8 days) Caco-2, T84, Vero, and HT29. **k** Internalization of *C. difficile* spores R20291 strain preincubated with serum of different mammalian species in Caco-2 cells. **i–k** Representative of $n = 3$ independent experiments. Error bars indicate the mean ± S.E.M. Statistical analysis was performed by two-tailed unpaired Student's *t* test, ns, $p > 0.05$. Scale bars **a** 5 μm; **c, d**, 100 nm; **e**, 1 μm; **f**, 2 μm; **g**, 200 nm; **h**, 500 nm.

***C. difficile* spore entry into intestinal epithelial cells requires Fn and Vn.** Fn and Vn are extracellular matrix proteins, which are also present in mammal serum and are widely used by enteric pathogens to infect host cells[16,17]. We have shown previously that both, Fn and Vn, bind in a concentration-dependent manner to *C. difficile* spores[15]. To assess whether serum Fn and Vn contribute to *C. difficile* spore entry, we evaluated the internalization assay in the presence of RGD peptide to block the interaction of Fn and Vn with their cognate receptors through the

RGD-binding domain[20,21]. RGD significantly reduced the extent of spore entry into differentiated Caco-2 cells in the presence of human serum by ~45% (Fig. 3a and Supplementary Fig. 8a), indicating that serum Fn and Vn might be involved in *C. difficile* spore entry in an RGD-specific manner; by contrast, no decrease in adherence of *C. difficile* spores was evidenced in the presence of RGD (Fig. 3b and Supplementary Fig. 8b). Similar results were observed in undifferentiated Caco-2 cells (Supplementary Fig. 8c–f). We confirmed these results by showing that the

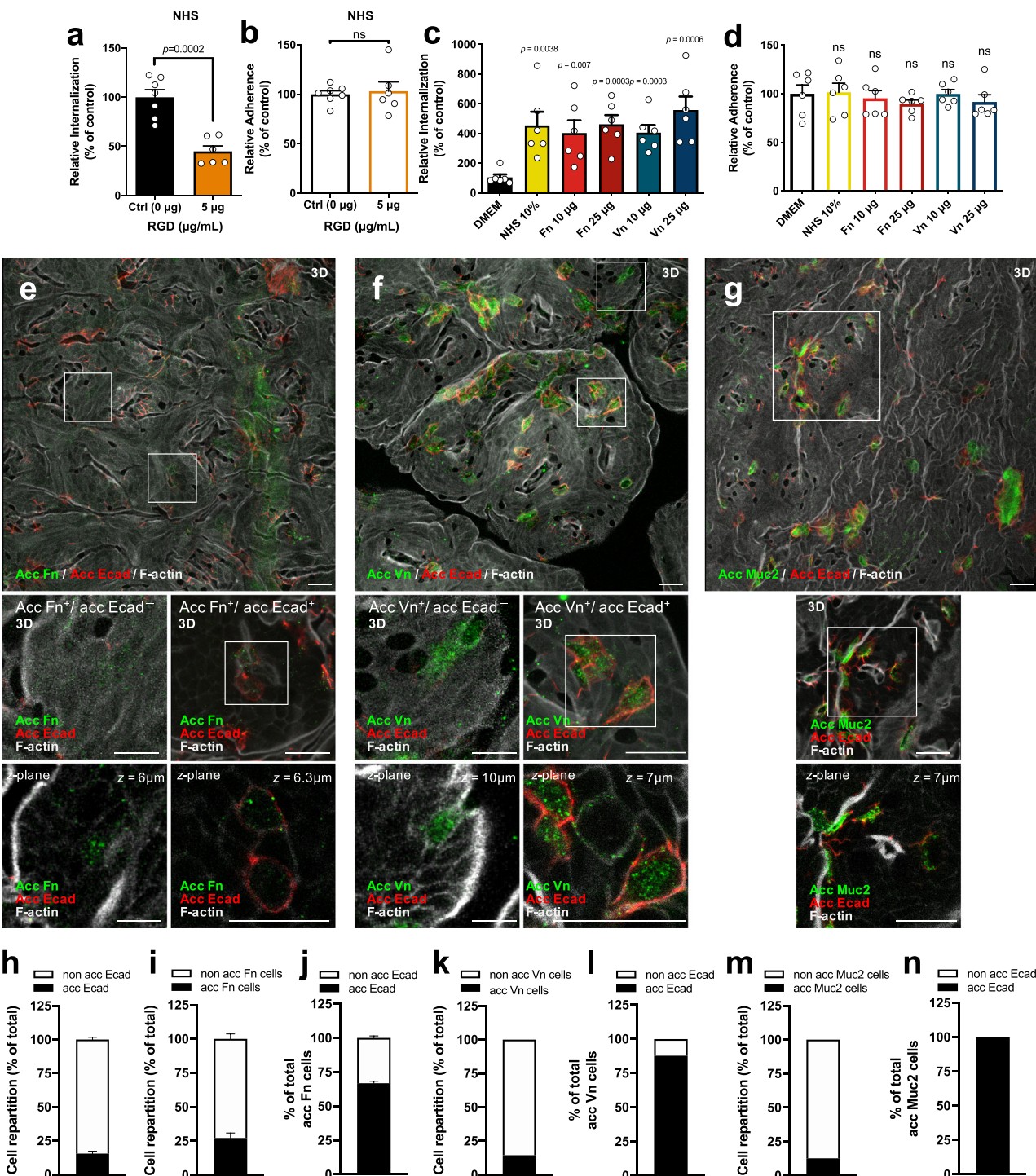

**Fig. 3 *C. difficile* spore internalization requires Fn and Vn which are luminally accessible in the intestinal barrier. a** Internalization **b** and adherence of *C. difficile* spores preincubated with NHS in differentiated monolayers of Caco-2 in the presence of RGD peptide. **c, d** Differentiated Caco-2 cells were infected with *C. difficile* spores preincubated with DMEM, NHS, Fn, or Vn. Data shown in each panel are normalized to the control. In bars, each dot represents one independent well from **a** to **d** three independent experiments. **e–g** Representative confocal micrographs of fixed whole-mount of the healthy colon of $n = 2$ mice for **e** accessible Fn (acc Fn); **f** accessible Vn (acc Vn); and **g** accessible Muc2 (acc Muc2) with accessible Ecad (acc Ecad). The main figure shown a 3D projection, below magnifications, and a z-stack of representative cells with different immunostaining. The figures **h–n** shows the cell repartition of cell immunodetected for **h** acc Ecad, **i** acc Fn, **j** total acc Fn cells that were immunodetected for acc Ecad, **k** cells immunodetected for acc Vn, **l** total acc Vn cells that were immunodetected for acc Ecad, **m** acc Muc2, **n** total acc Muc2 cells that were immunodetected for acc Ecad. **h–n** 1000–1200 cells were counted per field in two independent mice. **h** Four fields per mice, $n = 2$ mice; **i, j** two fields per mice, $n = 2$ mice; **k, l** one field per mice, $n = 2$ mice. Acc Fn, acc Vn, and Muc2 are shown in green, acc Ecad is shown in red, and F-actin in gray (fluorophores colors were digitally reassigned for a better representation). Scale bar, 20 μm. Error bars indicate mean ± S.E.M. Statistical analysis was performed by two-tailed unpaired Student's *t* test, ns, $p > 0.05$; ****$p < 0.0001$. Scale bar, 20 μm.

infection with spores preincubated with Fn or Vn restored spore entry into differentiated (Fig. 3c) and undifferentiated Caco-2 cells (Supplementary Fig. 9a), but had no impact on spore adherence to differentiated (Fig. 3d) and undifferentiated Caco-2 cells (Supplementary Fig. 9b). Similar results were observed in differentiated Caco-2 cells pretreated with Fn or Vn before infection with *C. difficile* spores (Supplementary Fig. 9c, d), confirming that the presence of Fn and Vn mediates *C. difficile* spore entry.

**Intestinal barrier sites with accessible Fn and Vn**. Both Fn and Vn are mainly located in the basal and basolateral membrane of epithelial cells, where tight and adherent junctions are formed[16,17]. However, several epithelial barrier suffers reorganization and/or disruption of tight and adherent junctions, such as cell extrusion sites, goblet cells (GCs), at cell–cell junctions with neighboring cells, and along villus epithelial folds[22–24]. Therefore, we hypothesized that sites undergoing adherent junction rearrangement also contained accessible Fn and Vn. We performed double staining, in which luminally accessible Fn and E-cadherin (Ecad), a marker for reorganization or disruption of adherent junctions[22], were stained in nonpermeabilized tissue. We first determined the relative number of IECs in the colonic tissue that expresses luminally accessible Ecad and found that nearly 16% of the IECs have this feature (Fig. 3h). Accessible Fn and Vn were observed in 27 and 14% of the IECs cells (Fig. 3i, k). We observed that most of the cells that had luminally accessible Fn or Vn also had accessible Ecad (Fig. 3j, l) and are likely undergoing a major reorganization of the adherent junctions. However, a small fraction of epithelial cells with accessible Fn (33%) or Vn (12%) had no accessible Ecad (Fig. 3i, k). Luminally accessible Ecad has been previously found around mucus-expelling GCs in mice intestinal tissue[22]. Therefore, to quantify the relative abundance of GCs with accessible Ecad in our experimental conditions, we performed double immunostaining for accessible Ecad and the GC-specific marker Muc2[22,25]. We observed that 13% of the IECs were positive for Muc2 in colonic tissue (Fig. 3m), and all of Muc2-positive cells were positive for accessible Ecad (Fig. 3n). Luminally accessible Ecad has also been observed in mice ileal tissue[22]; we observed that nearly 17% of the IECs of mice ileal had luminal accessible Ecad (Supplementary Fig. 10a, b). Next, we quantified the relative abundance of GCs in the ileum mucosa and observed that nearly 9% of total IECs were positive for Muc2 (Supplementary Fig. 10c), of which 70% were positives for accessible Ecad (Supplementary Fig. 10d). This data supports the notion that Fn and Vn are also accessible in the intestinal epithelial barrier. Altogether, these results demonstrate the existence of sites in the intestinal barrier that undergo a reorganization of adherent junctions that exhibit accessible Fn and Vn through which *C. difficile* spores can gain entry into the intestinal barrier.

***C. difficile* spores internalize via Fn-$\alpha_5\beta_1$ and Vn-$\alpha_v\beta_1$ integrin in vitro**. The Fn RGD loop between domains FnIII9 and FnIII10 enhances binding between Fn and $\alpha_5\beta_1$ integrin[16]; Vn also has a similar RGD loop that enhances binding to $\alpha_v\beta_1$ integrin[17]. To address whether binding of Fn and Vn to their cognate integrin receptors is required for *C. difficile* spore entry into IECs, monolayers of Caco-2 cells were infected with *C. difficile* R20291 spores in the presence of the inhibitory RGD peptide, showing that in the presence of Fn or Vn, increasing concentrations of RGD progressively decreased spore entry into differentiated (Fig. 4a, b) and into undifferentiated Caco-2 cells (Supplementary Fig. 11a, c), but not spore adherence (Fig. 4c, d and Supplementary Fig. 10b, d) to Caco-2 cells. Next, through an antibody blocking assay, we assessed which integrin subunits are

involved in Fn- and Vn-dependent entry of *C. difficile* spores into IECs. Results demonstrate that blocking the subunits of the collectin-binding, $\alpha_2\beta_1$ integrin[26,27] and $\beta_3$ integrin subunit did not affect internalization nor adherence of *C. difficile* spores to Caco-2 cells in the presence of Fn (Fig. 4e, f) or Vn (Fig. 4g, h). However, a significant decrease in spore entry, but not spore adherence, to differentiated and undifferentiated Caco-2 cells was observed upon blocking each subunit of $\alpha_5\beta_1$ integrin in the presence of Fn (Fig. 4e, f and Supplementary Fig. 11e, f), as well as blocking each subunit of $\alpha_v\beta_1$ integrin in the presence of Vn (Fig. 4g, h and Supplementary Fig. 11g, h). These results were confirmed upon expressing each integrin subunit in Chinese hamster ovary (CHO) cells (Fig. 4i–k), a naive cell line that otherwise does not express integrins. CHO cells expressing individual $\alpha_5$ or $\beta_1$ integrin subunits exhibited significant spore entry, but not adherence in the presence of Fn (Fig. 4l, m); equally, CHO cells expressing individual $\alpha_v$ or $\beta_1$ integrin subunit exhibited significant spore entry, but not spore adherence in the presence of Vn (Fig. 4n, o). No increase in entry or adherence was detected in the absence of Fn and Vn (Supplementary Fig. 12a, b). Altogether, these observations demonstrate that the internalization of *C. difficile* spores into IECs occurs through Fn-$\alpha_5\beta_1$ and Vn-$\alpha_v\beta_1$ uptake pathways.

**Fn and Vn bind to the hair-like extensions of *C. difficile* spores, formed by the collagen-like BclA3 exosporium protein**. *C. difficile* spores of epidemically relevant strains exhibit hair-like projections that are likely to be formed by the collagen-like exosporium proteins[1,13]. Fn and Vn have a gelatin/collagen-binding domain[16,17], suggesting that these molecules might interact with *C. difficile* spores through these hair-like projections. Indeed, through TEM coupled with immunogold labeling of Fn and Vn, we observed that more than ~50% of the spores were positive for Fn- or Vn-immunogold particles (Supplementary Fig. 13a, b); immunogold Fn- and Vn-specific particles were observed in proximity to the hair-like extensions of *C. difficile* R20291 spores (Fig. 5a, b), suggesting that these structures might be implicated in spore entry into IECs. Most epidemically relevant strains encode two collagen-like exosporium proteins, BclA2 and BclA3[1,13]. During the sporulation of R20291 strain, *bclA3* expression levels are ~60-fold higher than those of *bclA2*[28]. Consequently, we first hypothesized whether BclA3 was responsible for the formation of the hair-like extensions. Therefore, we constructed a single *bclA3* mutant strain, in an epidemic R20291 background, by removing the entire gene through a *pyrE*-based allelic exchange system[29] (Supplementary Fig. 14). Electron micrographs demonstrate that, as expected, wild-type R20291 ($\Delta pyrE/pyrE^+$) spores exhibited typical hair-like projections observed in previous reports[1,11,12] (Fig. 5c). By contrast, the $\Delta bclA3$ deletion mutant formed spores that lacked the hair-like projections (Fig. 5d) that were restored upon complementation of the $\Delta bclA3$ mutant strain with a single wild-type copy of *bclA3* in the *pyrE* locus ($\Delta bclA3/bclA3^+$; Fig. 5e), indicating that BclA3 is required for the formation of these projections on the surface of *C. difficile* spores.

**BclA3 is required for Fn-$\alpha_5\beta_1$- and Vn-$\alpha_v\beta_1$-mediated spore entry into IECs**. To address whether BclA3 exosporium protein is implicated in *C. difficile* spore entry into IECs, we first assayed whether the absence of BclA3 protein affected the internalization of *C. difficile* spores into IECs in the presence of Fn or Vn. As a control, we ensured that the anti-*C. difficile* spore goat serum used to quantify extracellular *C. difficile* spores, recognized $\Delta bclA3$ mutant spores (Supplementary Fig. 15a–c). Spores of the *C. difficile* $\Delta bclA3$ mutant strain exhibited a significant decrease

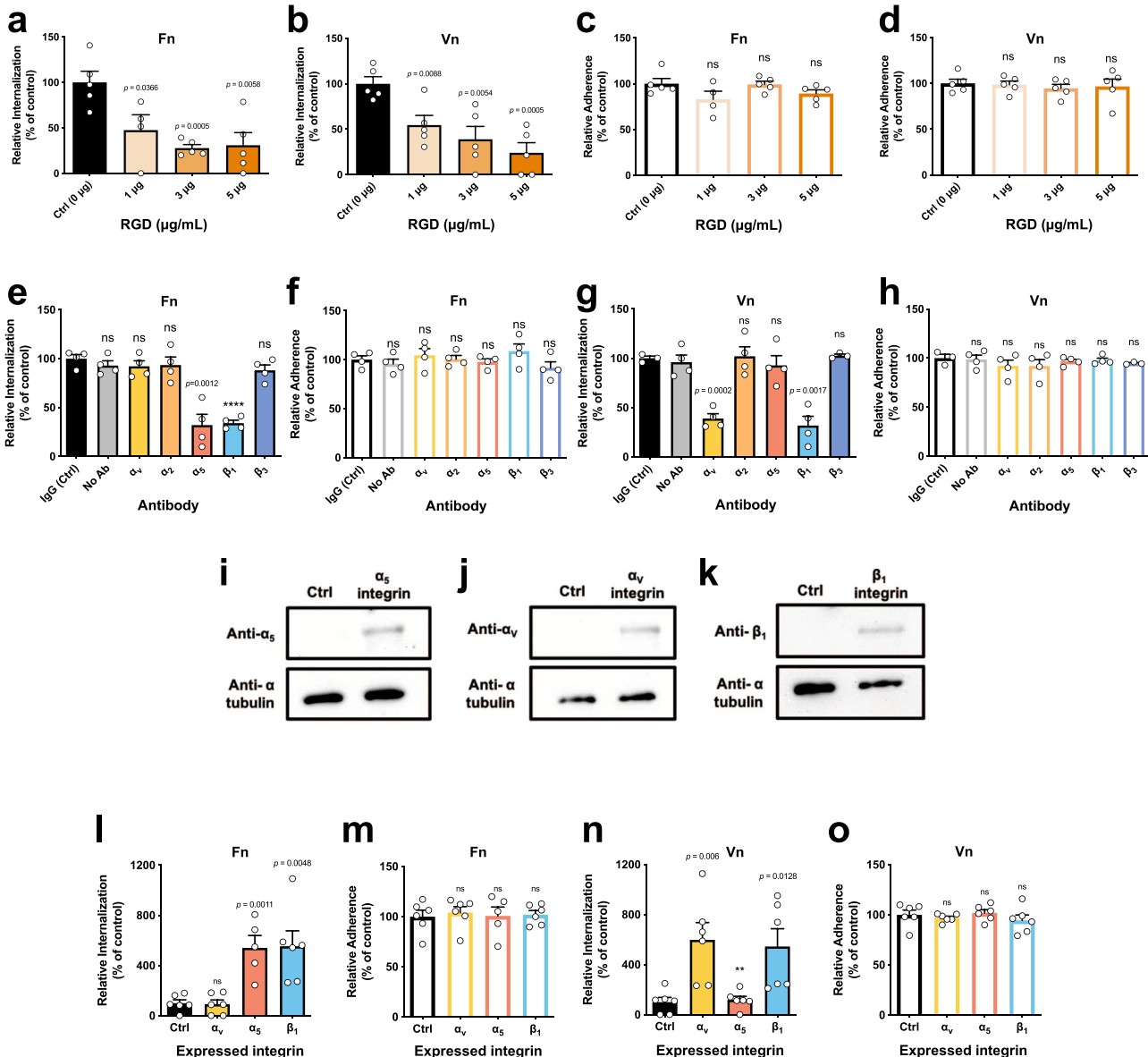

**Fig. 4 C. difficile spore internalization via fibronectin-α₅β₁ and vitronectin-αᵥβ₁ integrins intestinal epithelial cells.** *C. difficile* spore internalization in **a–d** differentiated Caco-2 cells incubated for 1 h with 1, 3, or 5 µg mL⁻¹ of RGD peptide and infected with *C. difficile* spores preincubated for 1 h with **a**, **c**, Fn and **b**, **d**, Vn. **e–h** Differentiated Caco-2 monolayers were incubated for 1 h with antibodies against αᵥ, α₂, α₅, β₁, β₃, nonimmune IgG antibody or without antibody. They were then infected with *C. difficile* spores R20291 preincubated with **e**, **f**, Fn or **g**, **h**, and Vn. **i–k** Immunoblotting of cell lysates of CHO cells transfected with ectopic expression of **i** α₅ (~120 kDa); **j** αᵥ (~120 kDa); and **k** β₁ (~120 kDa) and alpha tubulin as a loading control (50 kDa). *C. difficile* spore **l**, **n** internalization or **m**, **o** adherence in CHO cells ectopically expressing αᵥ, α₅, β₁ integrins, of spores pretreated 1 h with **l**, **o** Fn or **n**, **o** of Vn. Data shown in each panel are normalized to the control. In bars, each dot represents one independent well from three independent experiments. Error bars indicate mean ± S.E.M. Statistical analysis was performed by two-tailed unpaired Student's *t* test, ns, *p* > 0.05; **\*\****p* < 0.01; \*\*\*\**p* < 0.0001.

in spore entry into Caco-2 cells, but not adherence to monolayers of Caco-2 cells was observed upon infection with *C. difficile* spores Δ*bclA3* mutant in the presence of Fn (Fig. 5f, g) and Vn (Fig. 5h, i). Importantly, the defect in spore entry of the Δ*bclA3* mutant strain in the presence of Fn or Vn was restored to wild-type levels of internalization Δ*bclA3*/*bclA3*⁺ strain (Fig. 5f–i), indicating that BclA3 is required for Fn- and Vn-mediated internalization into IECs. We further confirmed these results in monolayers of HeLa cells, evidencing essentially identical results (Supplementary Fig. 16a–d). Next, to address whether Fn-α₅β₁ and Vn-αᵥβ₁-mediated spore entry is BclA3 specific, we carried out infection experiments with Δ*bclA3* mutant spores in

monolayers of CHO cells expressing individual integrin subunits. In the presence of Fn, a significant decrease in spore entry (Fig. 5j, k), but not in adherence (Supplementary Fig. 16e, f), was observed upon infection of CHO cells expressing the α₅ or β₁ integrin subunit with Δ*bclA3* mutant spores. Similarly, Δ*bclA3* mutant spores internalized to a significantly lesser extent than wild-type spores during infection of CHO cells expressing αᵥ or β₁ integrin receptors in the presence of Vn (Fig. 5l, m); however, the absence of BclA3 had no impact on spore adherence to CHO cells in the presence of Vn (Supplementary Fig. 16g, h). Fn- and Vn-mediated internalization of *C. difficile* spores into CHO cells expressing each integrin subunit was restored to wild-type

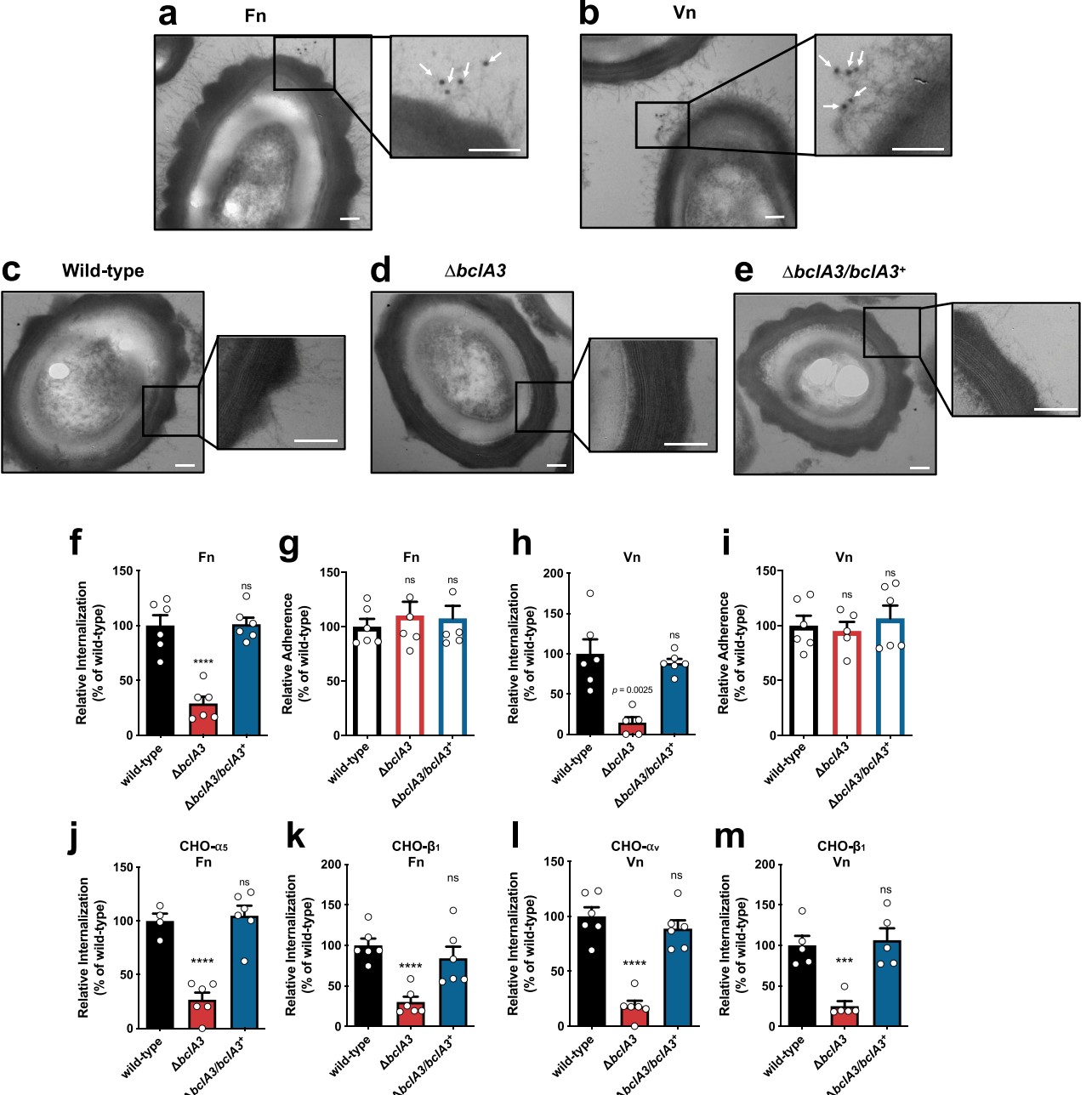

**Fig. 5 The collagen-like exosporium protein BclA3 is required for spore entry into intestinal epithelial cells via Fn-$\alpha_5\beta_1$ and Vn-$\alpha_v\beta_1$. a, b** *C. difficile* spores R20291 were incubated with Fn or Vn for 1 h. Samples were processed and visualized for TEM. White arrows indicate anti-Fn or -Vn rabbit antibody and anti-rabbit-gold 12 nm antibody complex. **c–e** TEM of wild type ($\Delta pyrE/pyrE^+$), $\Delta bclA3$, and $\Delta bclA3/bclA3^+$ *C. difficile* R20291 spores. **f, h** Internalization and **g, i** adherence in differentiated Caco-2 cells of wild type, $\Delta bclA3$, and $\Delta bclA3/bclA3^+$ *C. difficile* spores preincubated with **f, g** Fn or **h, i** Vn. **j, l** Internalization and **k, m** adherence in CHO cells ectopically expressing: **j** $\alpha_5$; **k, m** $\beta_1$; or **l** $\alpha_v$ infected with wild type, $\Delta bclA3$, and $\Delta bclA3/bclA3^+$ *C. difficile* spores preincubated with **j, k** Fn or **l, m** Vn. Data shows internalization and adherence normalized to wild-type spores. In bars, each dot represents one independent well from three independent experiments. Error bars indicate the mean ± S.E.M. Statistical analysis was performed by two-tailed unpaired Student's *t* test, ns, $p > 0.05$; ***$p < 0.001$; ****$p < 0.0001$. **a–e** Scale bar, 100 nm.

levels upon infection with spores $\Delta bclA3/bclA3^+$ (Fig. 5j–m). These results collectively demonstrate that BclA3-Fn-$\alpha_5\beta_1$ and BclA3-Vn-$\alpha_v\beta_1$ are two pathways through which *C. difficile* spores can internalize into non-phagocytic cells.

**Inactivation of the exosporium protein BclA3 decreases spore adherence, but not spore entry, of *C. difficile* spores to the intestinal mucosa.** To assess whether the collagen-like BclA3

exosporium protein also contributed to the internalization of *C. difficile* spores into the intestinal mucosa in vivo, we used a colonic and ileal loop mouse model (Fig. 6a–c and Supplementary Fig. 17a–c). In contrast to our in vitro data, analysis of colonic mucosa sections show that inactivation of *bclA3* leads to a significant decrease of ~60% in spore adherence per $10^5$ μm² to the ileum mucosa (Fig. 6d); however, no differences were observed in spore internalization relative to the total adhered spores (Fig. 6e).

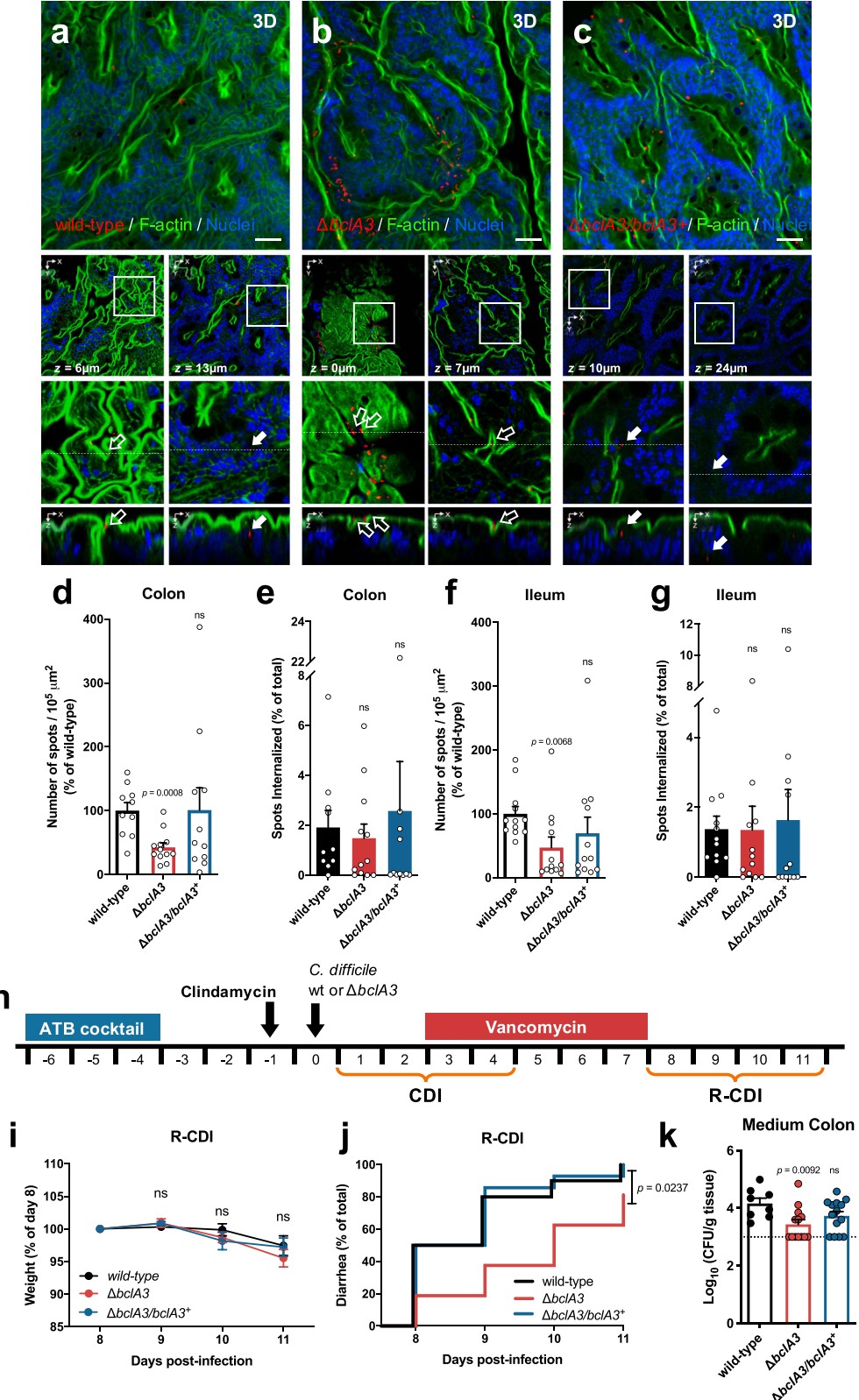

A similar trend was evidenced in ileal loops, where Δ*bclA3* spores adhered in a ~50% lower than wild-type spores per $10^5$ μm² (Fig. 6f); however, no differences were observed in spore internalization relative to total adhered spores (Fig. 6g). The defects in spore adherence to the colonic and ileum mucosa were restored to wild-type levels upon complementing the wild-type *bclA3* allele

in the Δ*bclA3* mutant (Fig. 6d–g and Supplementary Fig. 17a–c). Strikingly, these data indicate that the collagen-like BclA3 exosporium protein is required for *C. difficile* spore adherence to the intestinal mucosa and that additional spore-surface proteins or uncharacterized host factors are contributing to spore entry pathways in vivo.

**Fig. 6 BclA3 is involved in *C. difficile* spore adherence to the intestinal mucosa and delays the onset of diarrhea during R-CDI. a–g** Intestinal loops of approximately ~1.5 cm of the ileum and colon were injected with $5 \times 10^8$ *C. difficile* spores wild type (ileum $n = 12$; colon $n = 10$), $\Delta bclA3$ (ileum $n = 12$; colon $n = 12$), and $\Delta bclA3/bclA3^+$ (ileum $n = 12$; colon $n = 11$). **a–c** Representative confocal micrographs. *C. difficile* spores are shown in red, F-actin is shown in green, and nuclei in blue (fluorophores colors were digitally reassigned for a better representation). The white arrows and empty arrows denote internalized and adhered *C. difficile* spores. Number of the spots (spores) per $10^5$ µm$^2$ relatives to wild type of **d** adhered and the percentage of internalized spores in the **d**, **e** ileum, and in the **f**, **g** colonic mucosa, respectively. **h** Schematics of the experimental design. Mice were infected with $5 \times 10^7$ *C. difficile* spores strain, wild type ($n = 10$), $\Delta bclA3$ ($n = 16$), or $\Delta bclA3/bclA3^+$ ($n = 14$) and were treated with vancomycin from days 3 to 7 and were monitored daily for **i** relative weight during and **j** onset of diarrhea during the R-CDI. **k** Spore adherence to the medium colon was evaluated on day 11. Error bars indicate the mean ± S.E.M. Statistical analysis was performed by **d–g**, **k** by two-tailed Mann–Whitney test; **i** Kruskal–Wallis test, Dunn multiple comparison, **j** log-rank (Mantel–Cox) test; ns, $p > 0.05$.

**The *C. difficile* collagen-like BclA3 exosporium protein contributes to spore persistence and recurrence of the disease in mice.** Since BclA3 is essential for spore entry into IECs in vitro and for adherence in the intestinal mucosa in vivo, we hypothesized that BclA3 might mediate spore persistence and contribute to R-CDI. Therefore, antibiotic-treated mice were infected with spores of wild type ($\Delta pyrE/pyrE^+$), $\Delta bclA3$, and $\Delta bclA3/bclA3^+$ strains (Fig. 6h). All three groups of mice exhibited similar weight loss during the initiation of CDI, and all manifested signs of diarrhea within 3 days post infection (Supplementary Fig. 17d, e). Similar levels of *C. difficile* spores shed in feces were observed during the initiation of CDI between mice infected with wild type ($\Delta pyrE/pyrE^+$), and the mutants $\Delta bclA3$ and $\Delta bclA3/bclA3^+$ (Supplementary Fig. 17f). These results indicate that the absence of BclA3 does not affect the initiation of CDI. Next, the impact of BclA3 in the recurrence of the infection was assessed by treating *C. difficile*-infected mice with vancomycin for 5 days (Fig. 6h), R-CDI was monitored from day 8 post infection. No significant differences in weight loss were evidenced after vancomycin treatment (days 8–11; Fig. 6i). However, a significant delay in the onset of diarrhea during R-CDI was observed after vancomycin treatment in mice infected with $\Delta bclA3$ mutant spores compared to wild-type-infected mice (Fig. 6j). R-CDI defect observed in $\Delta bclA3$ mutant-infected mice was restored to wild-type levels with the complemented $\Delta bclA3/bclA3^+$ strain (Fig. 6j). Although there were no significant differences in the levels of *C. difficile* spores shed in the feces during R-CDI (Supplementary Fig. 17f), significantly lower CFUs of $\Delta bclA3$ mutant spores were detected in the medium colon compared to wild-type spores (Fig. 6k), but not in other sections of the intestinal tract (Supplementary Fig. 17g–i). Again, this defect was reverted in the complemented $\Delta bclA3/bclA3^+$ strain (Fig. 6k). Cytotoxicity levels in the cecum content of $\Delta bclA3$ mutant-infected mice were similar to those found in animals infected with wild-type strain (Supplementary Fig. 17j). These results collectively demonstrate that the collagen-like BclA3 exosporium protein is involved in *C. difficile* persistence and recurrence of the disease.

**Inhibition of spore entry into IECs renders *C. difficile* spores susceptible to taurocholate germination.** Since *C. difficile* spore entry into IECs requires integrin receptors, we tested whether cholesterol–lipid rafts, commonly required by integrin receptors for endocytosis[30,31], were also required for uptake of *C. difficile* by IECs. Therefore, we used the cholesterol-chelating agent, nystatin, a caveolin-related pathway inhibitor that disrupts membrane microdomains known to be implicated in integrin-mediated endocytosis and pathogen uptake[30,32]. Cells were preincubated with nystatin for 1 h at 37 °C and infected in the same medium containing the inhibitor and *C. difficile* spores. *C. difficile* spore entry was inhibited in a dose-dependent manner into Caco-2 cells and T84 cells (Fig. 7a, b

and Supplementary Fig. 18a, b). A total of 30 µM of nystatin inhibited the spore entry by ~80% in human cell lines Caco-2 and ~65% in T84 cells. We determined cell viability in the presence of nystatin by MTT at the highest concentration of the inhibitor. Cell viability was generally ~90% (Supplementary Fig. 18c). These results suggest that *C. difficile* spore entry is sensitive to cholesterol-sequestering compounds.

Vancomycin administration leads to increased fecal concentration of primary bile acids[33] leading to enhanced *C. difficile* spore germination[34,35], suggesting that luminal taurocholate would trigger germination of extracellular *C. difficile* spores that could subsequently become inactivated by vancomycin. Therefore, we hypothesize that intracellular spores should remain dormant in the presence of taurocholate. To test this hypothesis, monolayers of 1-h nystatin-treated or untreated Caco-2 cells were infected for 3 h with serum-treated *C. difficile* spores. Next, infected monolayers were washed and treated with taurocholate to trigger germination of extracellular spores, followed by ethanol treatment to inactivate germinated *C. difficile* spores. We observed that not all of the spores became ethanol-sensitive upon taurocholate treatment of infected Caco-2 monolayers (Fig. 7c), suggesting that internalized spores were protected from taurocholate-triggered germination. We confirmed this by evidencing a significant increase in ethanol-sensitive germinated spores in the presence of nystatin (Fig. 7c). These results indicate that blocking *C. difficile* spore entry contributes to taurocholate-triggered germination of *C. difficile* spores and subsequent spore inactivation.

**Inhibition of *C. difficile* spore entry into the intestinal barrier reduces recurrence of the disease in mice.** To address whether in vivo *C. difficile* spore entry into the intestinal barrier also required RGD-binding integrins[36], colonic and ileal loop assays were assessed in the presence of RGD during *C. difficile* spore infection (Fig. 7d, e and Supplementary Fig 18d, e). Ileal and colonic loops were injected with RGD peptide and *C. difficile* spores for 5 h, then were processed and visualized in confocal microscopy. Consistent with our in vitro data, in the colonic loop sections, we observed that presence of RGD peptide reduced spore internalization by ~82% (Fig. 7g), while no difference in spore adherence was observed (Fig. 7h); similarly, in ileal loop sections, we observed that RGD peptide decreased spore internalization by ~90% (Fig. 7i) and did not affect the spore adherence to the ileum mucosa (Fig. 7j). These results demonstrate that *C. difficile* spore entry in vivo is RGD-binding integrin-dependent.

Since the RGD dependency of spore entry into the intestinal barrier is likely attributed to integrin receptors, we address whether the cholesterol-sequestering drug, nystatin, could block internalization of *C. difficile* spores into the intestinal barrier in vivo in the colonic and ileum mouse mucosa. Mice were treated for 24 h with nystatin or saline as a control before surgery

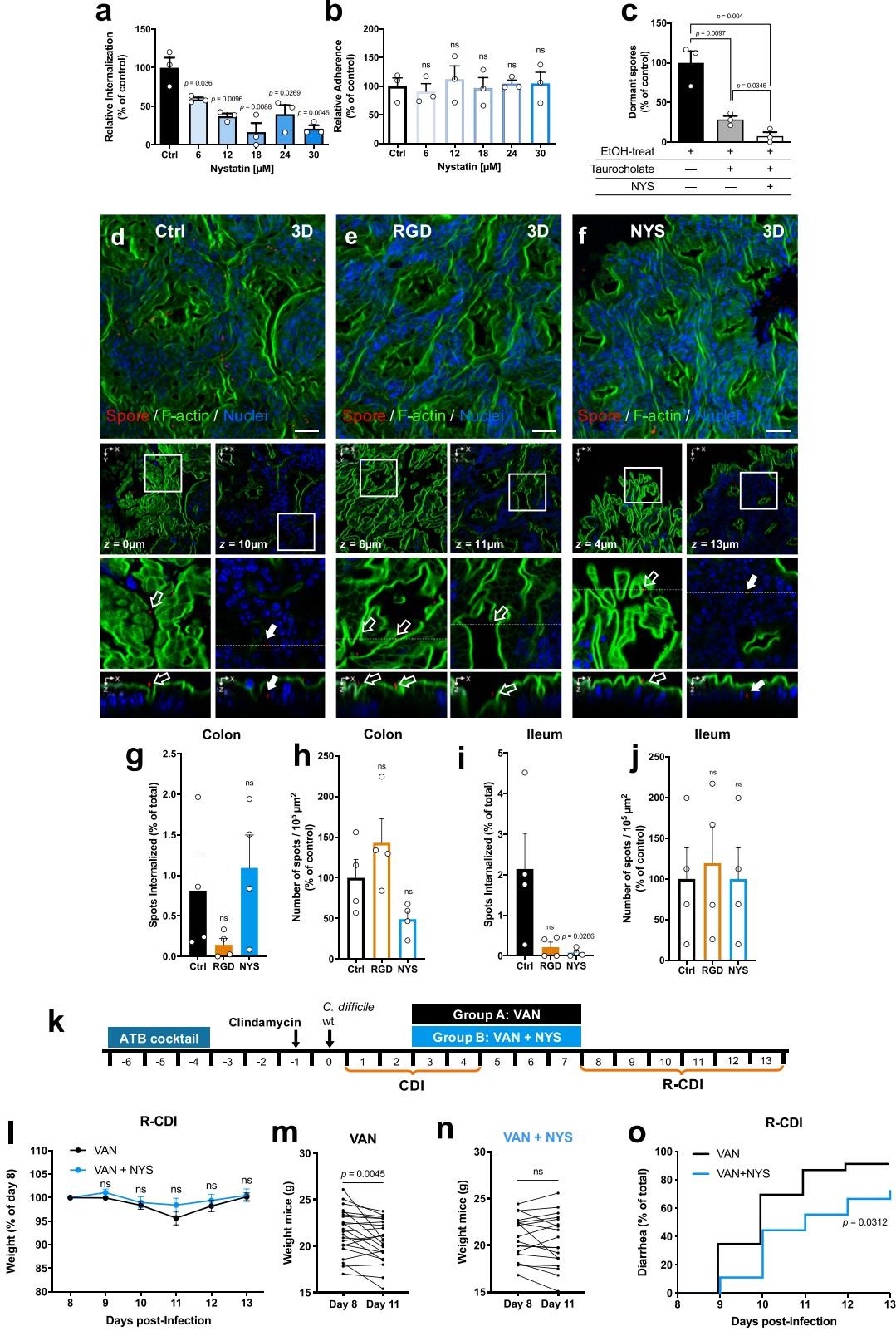

and during intestinal loop infection (Fig. 7d, f and Supplementary Fig 18d, f), then were infected with *C. difficile* spores for 5 h, then tissues were processed for confocal microscopy. In the colonic loop section, we observed that nystatin had no effect on spore internalization (Fig. 7g) and in spore adherence to the colonic mucosa (Fig. 7h); however, in the ileal loop sections, the presence

of nystatin significantly decreased spore internalization by ~96% (Fig. 7i), and no effect in *C. difficile* spore adherence to the ileum mucosa was observed (Fig. 7j).

Since *C. difficile* spore entry prevents taurocholate germination, contributing to the persistence of *C. difficile* spores during the disease; we hypothesized that administration of the inhibitor of

**Fig. 7 Nystatin reduces *C. difficile* spore internalization and reduces R-CDI rates. a** Internalization and **b** adherence of undifferentiated Caco-2 cells were pretreated nystatin for 1 h, and subsequently infected with *C. difficile* spores R20291 preincubated for 1 h with FBS. **c** Colony-forming units of spores on BHIS-CC with 0.1% of sodium taurocholate of a lysate of undifferentiated Caco-2 were pretreated with 30 μM of nystatin (shown as NYS), or DMEM alone as control and infected with *C. difficile* spores, and treated with taurocholate and ethanol (EtOH). Cells were lysed and plated, so only spores that remain dormant after taurocholate treatment (dormant) germinate in the plates. The number of CFU mL$^{-1}$ was determined, and the percentage of adherence relative to the control. **a–c** In bars, each dot represents one independent well representative of three independent experiments. Loops of approximately of the ileum and colon of C57BL/6 were injected with $3 \times 10^8$ *C. difficile* R20291 with RGD peptide ($n = 4$), or nystatin ($n = 4$) and saline as control ($n = 4$). **d–f** Representative confocal micrographs *C. difficile* spore is shown in red, F-actin is shown in green, and nuclei in blue (fluorophores colors were digitally reassigned for a better representation). The white arrow and empty arrow denote internalized and adhered *C. difficile* spores. Scale bar, 20 μm. Percentage of internalized spots (spores) and the number of adhered per $10^5$ μm$^2$ in the **g**, **h** colonic mucosa or in the **i**, **j** ileum mucosa. **k** Schematics of the experimental design of a mouse model of R-CDI. ATB cocktail treated mice were infected with $6 \times 10^7$ R20291 spores. The CDI symptoms were treated from days 3 to 7 with nystatin and vancomycin (shown as VAN + NYS; $n = 18$ mice) or vancomycin alone as control (shown as VAN; $n = 23$ mice) and were monitored daily for **l** relative weight during the R-CDI. Weight loss comparison for animals treated with **m** vancomycin or **n** vancomycin and nystatin, and **o** onset to diarrhea during the R-CDI. Error bars indicate the mean ± S.E.M. Statistical analysis was performed by **a–c** two-tailed unpaired Student's *t* test; **g–j**, **l** two-tailed Mann–Whitney test; post-Dunn's; **m**, **n** one-tailed Wilcoxon matched-pairs signed-rank test; **o** log-rank (Mantel–Cox) test; ns, $p > 0.05$.

spore entry, nystatin, during CDI treatment with vancomycin, could reduce the recurrence of the infection in a previously developed mouse model of R-CDI[8] (Fig. 7k). To address this question, antibiotic-treated mice were infected with *C. difficile* R20291 spores. During the first episode of CDI, both groups of mice had similar weight loss, the timing of the onset of diarrhea, and shed similar amounts of *C. difficile* spores during the initiation of CDI (days 1–3; Supplementary Fig. 18g–i). On day 3 post-infection, animals were treated with vancomycin or a mixture of vancomycin and nystatin for 5 days (Fig. 7k). Vancomycin-treated mice exhibited a significant decrease in weight during R-CDI, which became highest at day 11 post-infection (fourth day after vancomycin treatment; Fig. 7k, m). By contrast, CDI animals treated with vancomycin and nystatin had no significant decrease in weight loss during the recurrence of the infection (Fig. 7k, n). These observations were confirmed upon monitoring the onset of diarrhea during R-CDI (Fig. 7o); we observed a significant delay in the onset of recurrent diarrhea in CDI mice treated with the mixture of vancomycin and nystatin compared to vancomycin alone (Fig. 7o). The animals shed similar amounts of *C. difficile* spores during R-CDI (Supplementary Fig. 18i). Collectively, these results demonstrate that the administration of a pharmacological inhibitor of internalization of *C. difficile* spores during vancomycin treatment delays the incidence of recurrence of the infection.

## Discussion

During CDI, *C. difficile* spore formation is essential in the recurrence of the disease[7], yet the underlying mechanisms that correlate *C. difficile* spore persistence and recurrence of the disease remain unclear. In this study, we unravel a novel and unexpected mechanism employed by *C. difficile* spores to interact with the intestinal mucosa that contributes to the recurrence of disease. Our results have identified host molecules, cellular receptors, and a spore-surface ligand involved in spore entry into IECs. Notably, intracellular spores remain dormant in the presence of germinant. Using nystatin, a pharmacological inhibitor of spore entry in combination with antibiotic treatment leads to a reduction in the recurrence of the disease in mice. Together, these observations open a new angle for therapeutic interventions of CDI to prevent the recurrence of the disease.

Our results identified host molecules and cellular receptors involved in the entry of *C. difficile* spores into IECs. The presence of Fn or Vn allows *C. difficile* spores to gain intracellular access to IECs, in an RGD-specific manner and through specific integrin receptors (i.e., $\alpha_5\beta_1$ and $\alpha_v\beta_1$). These observations were

confirmed by the in vivo inhibition of *C. difficile* spore entry in the presence of the RGD peptide, which inhibits specifically interactions between Fn-$\alpha_5\beta_1$ and Vn-$\alpha_v\beta_1$[20,21]. Although Fn and Vn are mainly located in the basal and basolateral membrane of IECs, contributing to cell polarity[16,17], antibody staining of healthy ileum and colonic tissue demonstrates that Fn and Vn are luminally accessible in a significant fraction of the IECs. Most of these cells were positive for luminally accessible Ecad and suggests that these cell types include cell extrusion, cells next to extrusion sites, and epithelial folds that typically undergo adherent junction reorganization[22–24]. However, a small fraction of cells positive for luminally accessible Fn and Vn were negative for luminally accessible Ecad, suggesting a novel phenotype within cells at the intestinal epithelial barrier. We also confirmed previous observations in mice that identified GCs have luminally accessible Ecad[22], suggesting that *C. difficile* spores might also target these cell types to gain entry into the epithelial barrier. M cells are an additional cell type that might contribute to *C. difficile* spore entry into the intestinal epithelial barrier includes since they express $\beta_1$-integrin at the apical surface in contrast to its normal basolateral location in enterocytes[22,37,38]. The fact that in vivo spore entry was RGD-binding integrin-specific suggests that Fn and Vn are accessible and employed by *C. difficile* spores to gain entry into IECs, which is consistent with the presence of accessible Fn and Vn in ileal and colonic loops. It is noteworthy that while RGD-specific entry was observed in both ileal and colonic loops, nystatin was only able to reduce spore entry into the ileum, but not colonic mucosa. This suggests that caveolae-independent endocytosis of *C. difficile* spores might prime in the colonic epithelia. During CDI, *C. difficile* toxins disrupt adherent junctions, leading to progressive exposure of deep regions of the colonic epithelium as infection advances. One consequence of this cellular disorganization may alter the distribution of cell receptors that may lead to increased adherence and internalization of *C. difficile* spores into the intestinal mucosa. Together, these observations prompt further studies to address how epithelium remodeling contributes to persistence of *C. difficile* spores and recurrence of the disease.

Another major contribution of this work is the role of the spore-surface collagen-like BclA3 exosporium protein in *C. difficile* spore entry into IECs in a Fn-$\alpha_5\beta_1$- and Vn-$\alpha_v\beta_1$-dependent manner. Our previous work shows that Fn and Vn bind in a dose-dependent manner to *C. difficile* spores[15]. By immunogold-electron microscopy, our results demonstrate that Fn and Vn bind to the hair-like projections of *C. difficile* spores. We also demonstrate that they are formed by the collagen-like exosporium

glycoprotein BclA3. It is noteworthy that experiments with monolayers of Caco-2 cells and CHO cells expressing integrin subunits demonstrate that BclA3 is essential for spore entry in the presence of Fn and Vn in an integrin-dependent manner; results that contrast with BclA3 being essential for adherence to the intestinal mucosa, but not for spore entry into the intestinal barrier. Coupling these results with those of in vivo RGD-specific *C. difficile* spore entry into the intestinal barrier indicates that additional spore-surface proteins might play redundant roles during in vivo spore entry or some uncharacterized host factors may also contribute. Regardless of these incongruencies, we observed that BclA3 contributes to the recurrence of the disease in a mouse model, suggesting that BclA3-mediated spore adherence to the intestinal mucosa might contribute to spore persistence and recurrence of the disease. The differences in spore adherence to the colonic tissue after R-CDI observed in the medium colonic tissue of mice might relate to the absence of mucosal folds typically observed in the distal and proximal colon of mice[39]. Here, we have shown that BclA3 uses Fn and Vn, and their specific integrins to gain entry into IECs and that BclA3 is essential for *C. difficile* spore adherence to the intestinal mucosa and contributes to the recurrence of the disease.

The work presented here also shows that *C. difficile* spore entry into IECs contributes to spore dormancy in the presence of primary bile salts (i.e., taurocholate) and that blocking in vivo spore entry during antibiotic treatment (vancomycin) leads to reduced R-CDI in mice. This brings a broader understanding of how strict anaerobic spore formers can persist in the host and remain dormant in a dysbiosis environment enriched with bile acids that trigger spore germination. Intracellular bacterial spores may survive until released back to the luminal environment to recolonize the host. Although the precise mechanism of how intracellular spores would contribute to the recurrence of the disease is unclear and prompts further studies, it may involve the rapid renewal of the intestinal epithelium, which, due to rapid proliferation and differentiation of multipotential stem cells located in the crypts of Lieberkühn[40–43], renew the epithelial barrier every 5 days. The factors that contribute to infection recurrence, although partly linked to continued disruption of the microbiota[44], are also directly linked to the persistence of *C. difficile* spores in the host. This is particularly relevant for CDI, considering that the rates for recurrent CDI are around ~18–32% and may rise between 45 and 65% during subsequent recurrent episodes[2,44]. Importantly, this *C. difficile* spore entry phenotype provides an additional point of intervention of disease recurrence and therapeutic susceptibility. The cholesterol-sequestering drug, nystatin, is FDA approved for oral administration[45], raising new approaches to develop pharmacological formulations that target *C. difficile* spore entry during disease. Similarly, BclA3 and $\alpha_5\beta_1$ and $\alpha_v\beta_1$ integrins are also candidates drug targets to combat recurrent CDI.

## Methods

**Data reporting**. Sample size was not predetermined using statistical methods. For cell culture experiments, randomized wells were used. For animal experiments, mice were randomly assigned to the different groups. The investigators were not blinded the development of the animal experiments to avoid cross contamination between animals belonging to different treatment groups or infected with different strains. In the case of R-CDI experiments of mice treated with nystatin, we ensure the correct administration of the treatments to animals of the different groups.

**Bacterial strains and growth conditions**. *C. difficile* strains (see Supplementary Table 1) were routinely grown at 37 °C under anaerobic conditions in a Bactron III-2 anaerobic chamber (Shellab, USA) in BHIS medium: 3.7% weight vol$^{-1}$ brain heart infusion broth (BD, USA) supplemented with 0.5% weight vol$^{-1}$ yeast extract (BD, USA) and 0.1% weight vol$^{-1}$ L-cysteine (Merck, USA) or on BHIS agar plates. *Escherichia coli* strains were routinely grown aerobically at 37 °C under aerobic

conditions with shaking at $1 \times g$ in Luria-Bertani medium (BD, USA), supplemented with 25 µg mL$^{-1}$ chloramphenicol (Merck, USA), where appropriate.

For mutant construction, a defined *C. difficile* minimal medium (CDMM) media[46] was prepared and uracil-free medium when performing genetic selections. For CDMM broth preparation, 5× amino acids (50 mg mL$^{-1}$ casamino acids, 2.5 mg mL$^{-1}$ L-tryptophan, and 2.5 mg mL$^{-1}$ L-cysteine), 10× salts (50 mg mL$^{-1}$ Na$_2$HPO$_4$, 50 mg mL$^{-1}$ NaHCO$_3$, 9 mg mL$^{-1}$ KH$_2$PO$_4$, and 9 mg mL$^{-1}$ NaCl), 20× glucose (200 mg mL$^{-1}$ D-glucose), 50× trace salts (2.0 mg mL$^{-1}$ (NH$_4$)$_2$SO$_4$, 1.3 mg mL$^{-1}$ CaCl$_2$·2H$_2$O, 1.0 mg mL$^{-1}$ MgCl$_2$·6H$_2$O, 0.5 mg mL$^{-1}$ MnCl$_2$·4H$_2$O, and 0.05 mg mL$^{-1}$ CoCl$_2$·6H$_2$O), 100× iron (0.4 mg mL$^{-1}$ FeSO$_4$·7H$_2$O), and 100× vitamins (0.1 mg mL$^{-1}$ D-biotin 0.1, mg mL$^{-1}$ calcium-D-pantothenate, and 0.1 mg mL$^{-1}$ pyridoxine) stock solutions were made by dissolving their components in Milli-Q water and filter sterilizing (0.2-µm pore size) prior to use. Solutions were mixed to obtain a final CDMM media made of 10 mg mL$^{-1}$ casamino acids, 0.5 mg mL$^{-1}$ L-tryptophan, 0.5 mg mL$^{-1}$ L-cysteine, 5 mg mL$^{-1}$ Na$_2$HPO$_4$, 5 mg mL$^{-1}$ NaHCO$_3$, 0.9 mg mL$^{-1}$ KH$_2$PO$_4$, 0.9 mg mL$^{-1}$ NaCl, 10 mg mL$^{-1}$ D-glucose, 0.04 mg mL$^{-1}$ (NH$_4$)$_2$SO$_4$, 0.026 mg mL$^{-1}$ CaCl$_2$·2H$_2$O, 0.02 mg mL$^{-1}$ MgCl$_2$·6H$_2$O, 0.01 mg mL$^{-1}$ MnCl$_2$·4H$_2$O, 0.001 mg mL$^{-1}$ CoCl$_2$·6H$_2$O, 0.004 mg mL$^{-1}$ FeSO$_4$·7H$_2$O, 0.001 mg mL$^{-1}$ D-biotin, 0.001 mg mL$^{-1}$ calcium-D-pantothenate, and 0.001 mg mL$^{-1}$ pyridoxine[46]. For solid medium, agar (BD, USA) were mixed with CDMM to a final concentration of 1.0% weight vol$^{-1}$. Finally, media were supplemented with uracil (Sigma-Aldrich, USA) at 5 mg mL$^{-1}$ and 5-fluoroorotic acid (5-FOA; USBiological, USA) at 2 mg mL$^{-1}$ as described[47,48].

**Cell lines and reagents**. Caco-2, Vero, HT29, HeLa, and CHO were obtained from ATCC (USA). Dr. Mauricio Farfán (Universidad de Chile, Chile) gently provided T84 cells. Caco-2, Vero-E6, and HeLa were routinely grown at 37 °C with 5% of CO$_2$ with Dulbecco's modified Eagle's minimal essential medium (DMEM) high glucose (HyClone, USA); CHO cells in Ham's F-12K (Kaighn's) medium; T84 in DMEM/F12 1:1 (HyClone, USA); and HT29 in RPMI 1640. All media were supplemented with 10% vol vol$^{-1}$ inactivated FBS (HyClone, USA) and 100 U mL$^{-1}$ penicillin, and 100 µg mL$^{-1}$ streptomycin (HyClone, USA). T84 cells were cultured onto Transwell (Corning, USA) until 1000–2000 Ω. For transfected CHO cells (CHO-$\alpha_v$, CHO-$\alpha_5$, and CHO-$\beta_1$), the culture media was supplemented with 1500 µg mL$^{-1}$ geneticin (HyClone, USA). For immunofluorescence experiments, cells were plated over glass coverslip in a 24-wells plate and cultured for 2 days post confluence (undifferentiated) or 8 days post confluence (differentiated), changing the culture medium every other day.

**Spore preparation**. A total of 100 µL of 1:1000 dilution of an overnight culture in BHIS was plated in 70:30 agar plates that were prepared as follow: 6.3% weight vol$^{-1}$ bacto peptone (BD, USA), 0.35% weight vol$^{-1}$ proteose peptone (BD, USA), 0.07% ammonium sulfate (NH$_4$)$_2$SO$_4$ (Merck, USA), 0.106% weight vol$^{-1}$ Tris base (Omnipur, Germany), 1.11% weight vol$^{-1}$ brain heart infusion extract (BD, USA), and 0.15% weight vol$^{-1}$ yeast extract (BD, USA), 1.5% weight vol$^{-1}$ Bacto agar (BD, USA). Plates were incubated for 7 days at 37 °C under anaerobic conditions in anaerobic chamber Bactron III-2 (Shellab, USA). Then plates were removed from the chamber, and colonies were scraped out with ice-cold sterile Milli-Q water. Then the sporulated culture was washed five times with ice-cold Milli-Q water in microcentrifuge at 18,400 × g for 5 min each. The sporulated culture was loaded in 45% weight vol$^{-1}$ autoclaved Nycodenz (Axell, USA) solution and centrifuged at 18,400 × g for 40 min to separate spores. Spore pellet was separated and washed five times at 18,400 × g for 5 min with ice-cold sterile Milli-Q water to remove Nycodenz. Spores were counted in Neubauer chamber, and volume adjusts at $5 \times 10^9$ spores mL$^{-1}$ and stored at −80 °C.

**C. difficile mutant construction by allelic exchange**. Primer design and amplification of *C. difficile* R20291 strain were based on the available *C. difficile* genomes from the EMBL/GenBank databases with accession number FN545816. The oligonucleotides and the plasmids/strains used in this study are listed in Supplementary Tables 1 and 2, respectively. In-frame deletions in *C. difficile* R20291 were made by allelic exchange using *pyrE* alleles[47].

To remove the *bclA3* gene, a 1086 bp allelic exchange cassette was obtained by overlap extension PCR of the LHA and RHA originated by amplification with primer pairs P332 (FP-LHA-bclA3-pyrE)/P334 (RP-LHA-bclA3-pyrE) and P335 (FP-RHA-bclA3-pyrE)/P336 (RP-RHA-bclA3-pyrE), each of 544 and 542 bp in size. The resulting cassette yielded complete removal of the entire *bclA3* cassette. Next, this cassette was cloned into Sbf1/AscI sites in pMTL-YN4, giving plasmid pDP376. To verify the correct construction of the plasmids, all constructs were Sanger sequenced.

The plasmids obtained were transformed into *E. coli* CA434 (RP4) and mated with *C. difficile* R20291 Δ*pyrE*[47]. *C. difficile* transconjugants were selected by subculturing on BHIS agar containing 15 µg mL$^{-1}$ thiamphenicol (Sigma-Aldrich, USA) and 25 µg mL$^{-1}$ cefoxitin (Sigma-Aldrich, USA) and re-streaked five times. The single-crossover mutants identified were streaked onto CDMM[49] with 1.5% weight vol$^{-1}$ agar supplemented with 2 mg mL$^{-1}$ 5-FOA (USBiological, USA) and 5 µg mL$^{-1}$ uracil (Sigma-Aldrich, USA) to select for plasmid excision. Confirmation of plasmid excision was made by negative selection in BHIS-thiamphenicol plates. The isolated FOA-resistant and thiamphenicol-resistant colonies were screened using the primer pair P664 (FP-bclA3-detect)/P665

(RP-bclA3-detect) for the *bclA3* mutant. All mutants were whole-genome sequenced to confirm the genetic background and that no additional SNPs were introduced during the genetic manipulation. For correction of the *pyrE* mutation, transconjugants with pMTL-YN2C were streaked onto minimal media without uracil or FOA supplementation, and developed colonies were analyzed further.

**Complementation by allelic exchange at the *pyrE* locus.** To complement the Δ*bclA3* mutation, a 3564 bp fragment containing 372 bp upstream of the start codon of *bclA3*, and the entire bicistronic operon formed by *sgtA* and *bclA3* was PCR amplified with primer pairs P476 (NFP-bclA3c-promotor)/P477 (NRP-bclA3c) and cloned into BamHI/EcoRI sites of pMTL-YN2C, giving plasmid pMPG1. Next, plasmid pMPG1 was transformed into *E. coli* CA434 and subsequently conjugated with *C. difficile* R20291Δ*pyrE* Δ*bclA3*, respectively, as described above. The transconjugants obtained were streaked onto CDMM and tested by colony PCR using primer pair P530 (Fp-pyrE detect)/P529 (RP-pyrE detect) for *pyrE* reversion. Complemented strains were also subjected to whole-genome sequencing.

**Transfections of CHO cells with $\alpha_5$, $\alpha_v$, and $\beta_1$ integrins.** The integrins subunits were overexpressed in CHO cells line with the following plasmids: Alpha 5 integrin-GFP (Addgene plasmid # 15238)[50], miniSOG-Alpha-V-Integrin-25 (Addgene plasmid # 57763)[51], and Beta1-GFP in pHcgreen donated by Martin Humphries (Addgene plasmid # 69804)[52]. CHO cells were seeded on coverslips in 24-well plates until they reached 70–90% of confluency and were transfected using Lipofectamine® LTX (Invitrogen, USA) according to manufacturer protocol with 1 µg of each plasmid. Transfected cells were analyzed and confirmed by positive GFP fluorescence in epifluorescence microscopy. When the population GFP-positive cells were higher than 50%, were selected for geneticin resistance with 1500 µg mL$^{-1}$ of geneticin until ~100% of GFP-positive cells, and the level of expression of the integrin subunits in the cells was confirmed by Western blot.

**SDS–PAGE and western blot of transfected CHO cells.** Transfected CHO cells were washed and homogenized with RIPA buffer that was prepared as follow: 50 mM buffer Tris HCl (Omnipur, Germany); 150 mM NaCl (Sigma-Aldrich, USA); 0.5% weight vol$^{-1}$ deoxycholate (Sigma-Aldrich, USA); 1% vol vol$^{-1}$ NP 40 (Sigma-Aldrich, USA); 1 mM EGTA (Sigma-Aldrich, USA); 1 mM EDTA (Sigma-Aldrich, USA); and 0.1% weight vol$^{-1}$ SDS (Winkler, USA). The cell lysate was centrifuged at $18,400 \times g$ for 30 min at 4 °C, and protein concentration was quantified by BCA protein kit (RayBiotech, USA). Next, 20 µg of protein were suspended in 2× SDS–PAGE sample loading buffer, boiled, and electrophoresed on 12% vol vol$^{-1}$ and 4% vol vol$^{-1}$ acrylamide SDS–PAGE gels (Bio-Rad Laboratories, Canada) on MiniProtean® camera (Bio-Rad Laboratories, Canada). Then proteins were transferred to a nitrocellulose membrane (Bio-Rad Laboratories, Canada). Membranes were blocked then probed in Tris-buffered saline containing 0.1% vol vol$^{-1}$ Tween (TTBS), with 2% weight vol$^{-1}$ BSA, incubated with mouse anti-$\alpha_v$, $\alpha_5$, and $\beta_1$ antibody (SC166665, SC376156 y SC374429; Santa Cruz Biotechnologies, USA) and 1:1000 mouse anti-alpha tubulin (T5168 Sigma-Aldrich, USA) in 2% BSA-TTBS as loading control at concentration and were washed three times with TTBS. Membranes were incubated with 1:10,000 vol vol$^{-1}$ secondary antibody anti-mouse horseradish peroxidase (HRP) conjugate (A5278, Sigma-Aldrich, USA) in 2% weight vol$^{-1}$ BSA–TTBS. HRP activity was detected with a chemiluminescence detection system (Fotodyne Imaging system, USA.) using PicoMax sensitive chemiluminescence HRP substrate (Rockland Immunochemicals, USA.) For full scans of western blots, please see Source data file.

**Germination assay of extracellular *C. difficile* spores in Caco-2 cells.** Two-day old confluent monolayers of Caco-2 cells were treated with 30 µM of nystatin for 1 h at 37 °C or DMEM high glucose without FBS as control. To infect cells, *C. difficile* spores at an MOI of 10 were preincubated 1 h at 37 °C with 20 µL of NHS and then suspended in 200 µL that was added to each well; FBS final concentration 10% vol vol$^{-1}$.

To remove unbound spores, monolayers were washed three times with PBS, and cells were incubated with 0.1% weight vol$^{-1}$ sodium taurocholate (Sigma-Aldrich, USA) in DMEM for 1 h at 37 °C (or DMEM as control) and washed three times with PBS. Cells were treated with 100% ethanol for 10 min, and cells were lysed with PBS–0.06% Triton X-100 for 10 min, plated in BHIS-CC supplemented with 0.1% weight vol$^{-1}$ sodium taurocholate, and incubated at 37 °C overnight. The number of CFU mL$^{-1}$ was determined, and the percentage of adherence relative to the control.

**Infection of monolayers of cell lines with *C. difficile* spores.** To evaluate the dynamic of *C. difficile* spore in IECs, Caco-2, and T84 cells were grown on coverslips in 24-wells tissue culture plates until they reached a monolayer of 2 days post confluency and were infected for 0.5, 1, 3, 5, and 8 h at 37 °C at an MOI of 10 of *C. difficile* spores preincubated 1 h at 37 °C with 20 µL of FBS, and then suspended in the infection volume of 200 µL that was added to each well. Then were washed gently in PBS before immunostaining as described below.

To evaluate if *C. difficile* internalize in different cell lines, undifferentiated, differentiated Caco-2, T84, Vero, and HT29 cells were infected at 37 °C with an MOI 10 with *C. difficile* spore strain 630, and R20291 preincubated 1 h at 37 °C with FBS or DMEM as control, as was described above. Were washed gently in PBS before immunostaining, as described below.

Also, undifferentiated Caco-2 cells were infected at an MOI of 10 with *C. difficile* spores preincubated 1 h at 37 °C with FBS, mouse serum (Pacific Immunology, USA), rabbit serum (Pacific Immunology, USA), and NHS (Complement Technology, USA), as was described above. Then were washed gently in PBS before immunostaining, as described below.

To evaluate if *C. difficile* internalization occurs in different strains, undifferentiated Caco-2 cells were infected at an MOI of 10 with *C. difficile* spores R20291, M120, and spores of *C. difficile* clinical isolates PUC52, PUC30, PUC25, PUC31, PUC 98, and PUC 131[53], which were preincubated 1 h at 37 °C with FBS, as was described above. Then were washed gently in PBS before immunostaining, as described below.

To assess that the internalization of *C. difficile* spores into IECs is through the specific interaction between Fn or Vn and their cognate integrin receptors, infection experiments were done in the presence of the RGD peptide[20,21]. Briefly, differentiated and undifferentiated Caco-2 cells were incubated with 0, 1, 3, and 5 µg mL$^{-1}$ of RGD peptide (Abcam, USA) for 1 h, 37 °C then were infected for 3 h at 37 °C with spores preincubated 1 h at 37 °C with NHS, as was described above. Then, samples were washed gently in PBS before immunostaining, as described below.

To evaluate whether the internalization of the spores is mediated by Fn and Vn, differentiated and undifferentiated Caco-2 cells were treated for 1 h 37 °C at with 10 µg mL$^{-1}$ of purified human Fn or human Vn in DMEM, and then were infected with an MOI 10 of untreated *C. difficile* spores R20291. The infection was also performed using untreated Caco-2 cells that were infected for 3 h at 37 °C with an MOI 10 of *C. difficile* spores preincubated for 1 h at 37 °C with 10 µg mL$^{-1}$ of human Fn or human Vn in DMEM. Then samples were washed in PBS before the immunostaining, as described below.

To confirm that the internalization of *C. difficile* spores is dependent on Fn and Vn, we perform an infection assay in differentiated and undifferentiated Caco-2 cells that were preincubated with 1, 3, or 5 µg mL$^{-1}$ of RGD peptide and were infected with an MOI of 10 with *C. difficile* spores preincubated for 1 h 37 °C with 10 µg mL$^{-1}$ of purified human Fn or human Vn in DMEM. Then samples were washed in PBS before the immunostaining, as described below.

To identify the integrin subunits implicated in spore entry, an antibody blocking assay was performed using mouse monoclonal antibodies against individual integrin subunits: anti-human integrin $\alpha_5$ and $\alpha_v$, (ab78614, ab16821; Abcam, USA), $\alpha_2$, $\beta_1$, and $\beta_3$ (MAB1950Z, MAB1959Z, and MAB2023Z; Millipore, USA); and control nonimmune IgG antibody (I5006, Sigma-Aldrich, USA). Caco-2 cells were incubated with 200 µL of DMEM with the appropriate antibodies at 5 µg mL$^{-1}$ for 1 h at 37 °C. The cells were infected for 3 h at 37 °C with spores preincubated for 1 h at 37 °C with 10 µg mL$^{-1}$ of purified human Fn or human Vn in DMEM. Then samples were washed in PBS prior to the immunostaining, as described below.

In order to demonstrate that *C. difficile* spore entry requires the integrins subunits $\alpha_5$, $\alpha_v$, and $\beta_1$, then CHO cells with ectopic expression of $\alpha_5$, $\alpha_v$, and, $\beta_1$ integrins were infected 3 h at 37 °C with an MOI of 10 of *C. difficile* spores that were preincubated with 10 µg mL$^{-1}$ of purified human Fn or human Vn in DMEM. Then samples were washed in PBS before the immunostaining, as described below.

To evaluate whether the collagen-like exosporium protein BclA3 is required for *C. difficile* spore entry into IECs is dependent of differentiated Caco-2 cells and monolayers of HeLa cells, were infected for 3 h at 37 °C with an MOI of 10 with wild type (Δ*pyrE*/*pyrE*$^+$), Δ*bclA3*, and Δ*bclA3*/*bclA3*$^+$ *C. difficile* R20291 spores that were preincubated for 1 h at 37 °C with 10 µg mL$^{-1}$ of purified human Fn or human Vn in DMEM. Then samples were washed in PBS before immunostaining, as described below.

To evaluate the effect of nystatin in the internalization of *C. difficile* spores, Caco-2 cells were preincubated with 6, 12, 18, 24, and 30 µM nystatin or T84 cells were incubated with 30 µM of nystatin (Sigma-Aldrich, USA) for 1 h at 37 °C in DMEM, and in the same media were infected for 3 h at 37 °C with spores at an MOI 10 preincubated with FBS, as was described above. At the used concentration of nystatin, the cell viability of treated Caco-2 cells and T84 for 4 h was ~90%, as was observed by trypan blue (Invitrogen, USA) and MTT assay (Life Technologies, USA) according to manufacturer protocols.

**Immunofluorescence of adhered *C. difficile* spores in infected monolayers and epifluorescence analysis.** The aforementioned infected monolayers of cells were subsequently fixed with PBS–4% paraformaldehyde for 10 min. They were then washed three times and blocked with PBS–1% BSA overnight at 4 °C; in non-permeabilized monolayers, the extracellular spores were marked with 1:50 anti-*C. difficile* spore serum (recognize specifically to *C. difficile* spores in infections of IECs infection assays; PAC 5573 Pacific Immunology, USA) in PBS–1% BSA 1 h room temperature (RT) and 1:400 anti-goat conjugated with CFL 488 secondary antibody (green; SC362255, Santa Cruz Biotechnologies, USA) in PBS–1% BSA for 1 h RT. The samples were washed three times with PBS and once with sterile distilled water. Samples were then dried at RT for 30 min, and coverslips were mounted using Dako Fluorescence Mounting Medium (Dako, Denmark) and

sealed with nail polish. Samples were observed on an Olympus BX53 fluorescence microscope with UPLFLN 100× oil objective (numerical aperture 1.30). Images were captured with the microscope camera for fluorescence imaging Qimaging R6 Retiga and pictures were analyzed with ImageJ (NIH, USA). Extracellular spores or adhered were considered as spores in phase contrast that were marked in fluorescence. Internalized spores were considered as spores visible in phase contrast, but that does not have fluorescence. A total of ~300 spores were analyzed per experimental condition.

**Mice used**. A 6–8 weeks C57BL/6 (male or female) were obtained from the breeding colony at the Biological Science Department of Universidad Andrés Bello derived from Jackson Laboratories. Mice were housed with ad libitum access to food and water. Bedding and cages were autoclaved, and mice had a 12-h cycle of light and darkness. Mice were housed at 20–24 °C with 40–60% of humidity. All procedures complied with all relevant ethical regulations for animal testing and research. This study received ethical approval by the Institutional Animal Care and Use Committee of the Universidad Andrés Bello.

**Colonic and ileal loop assay**. C57BL/6 mice were anesthetized in an isoflurane chamber (RWD, USA) with 4% vol vol$^{-1}$ isoflurane (Baxter, USA) and were maintained with 2% vol vol$^{-1}$ during the surgery administrated by air. A midline laparotomy was performed, making 1-cm incision in the abdomen, 1.5 cm ileal, and proximal colon (at 1.0–1.5 cm from the cecum as a reference) were ligated with silk surgical suture[18]. No postsurgical analgesic was used. To evaluate the effect of nystatin and RGD peptide in *C. difficile* spore internalization in vivo, 24 h prior to surgery, mice (*n* = 4) were treated with nystatin (17,000 IU kg$^{-1}$) in 100 µL of DPBS by oral gavage; control mice (*n* = 8; 4 for control and 4 for RGD treatment) where treated with 100 µL of DPBS by oral gavage. On the day of surgery, ileal and colonic ligated loops of control mice (*n* = 4) were injected with 100 µL of DPBS containing 3 × 10$^8$ *C. difficile* R20291 spores; ileal and colonic ligated loops of nystatin-treated mice (*n* = 4) were injected with 100 µL of DPBS containing 3 × 10$^8$ *C. difficile* R20291 spores and 340 IU (17,000 IU kg$^{-1}$) nystatin; ileal and colonic ligated loops of RGD-treated mice (*n* = 4) were injected with 100 µL of DPBS containing 3 × 10$^8$ *C. difficile* R20291 spores and 86.6 µg (250 nM) of RGD peptide. DPBS was used to resuspend nystatin and RGD peptides because it rendered higher solubility than saline solution (0.9% weight vol$^{-1}$ NaCl). To evaluate the role of BclA3 protein in *C. difficile* spore internalization, the ligated loops were injected with 100 µL of 0.9% weight vol$^{-1}$ NaCl containing 5 × 10$^8$ wild-type spore (Δ*pyrE*/*pyrE*$^+$; ileum *n* = 12; colon *n* = 10), Δ*bclA3* (ileum *n* = 12; colon *n* = 12), and Δ*bclA3*/*bclA3*$^+$ (ileum *n* = 12; colon *n* = 11). The intestine was returned to the abdomen, and the incision was closed. Animals were allowed to regain consciousness. Mice were kept for 5 h and were euthanized. The ligated loops were removed and washed gently in PBS before immunostaining, as described below.

**Immunostaining of ileal and colonic loops**. First, extracted washed intestinal tissues from the ileal and colonic loops were longitudinally cut, then washed by immersion in PBS three times at RT. For better visualization of the tissues, they were fixed flat at RT. To perform this, tissues were fixed over a filter paper imbibed with 30% sucrose (Winkler, Chile) in PBS–4% paraformaldehyde (Merck, USA) for at least 15 min. Tissues were transferred to a microcentrifuge tube with the same fixing solution and were incubated at 4 °C overnight. Since mucus fixation with cross-linking agents, such as paraformaldehyde, cause mucus layer of colon to collapse and shrink to a very tiny lining, the epithelia[54], we did not observe mucus layer in our ileal and colonic loops. Before immunostaining, the intestinal and colonic tissues were cut into ~5 × 5 mm fragments.

To quantify *C. difficile* spore adherence and internalization in the colonic and in the ileal mucosa, tissues were made permeable by incubation with PBS–0.2% Triton X-100 (Merck, USA) and blocked with PBS–3% BSA (Sigma-Aldrich, USA) for 3 h at RT, the same buffer was used for subsequent incubation with antibodies. Tissue was incubated with a primary polyclonal 1:1000 anti-*C. difficile* spore IgY batch 7246 antibodies (Aveslab, USA) in PBS–3% BSA that does not immunoreacted with epitopes of vegetative cells neither with murine microbiota[8], and with 1:50 phalloidin Alexa-Fluor 568 (#A12380 Thermo Fisher, USA) in PBS–3% BSA overnight at 4 °C to stain the actin cytoskeleton. Following PBS washed, samples were incubated with 1:400 goat anti-chicken IgY secondary antibodies Alexa-Fluor 488 (#ab150173 Abcam, USA) in PBS–3% BSA at RT, washed three times with PBS, and the cellular nuclei stained with 1:1000 of Hoechst 33342 (ThermoFisher, USA) for 15 min at RT.

To perform double immunostaining accessible Fn, accessible Vn, accessible Muc2, with accessible Ecad in healthy colonic tissue or accessible Muc2 with accessible Ecad in the intestinal tissue, first, the proximal colon and ilium of two independent, healthy C57BL/6 mice of 8 weeks old were removed, and mice were sacrificed. Next, tissues were washed by immersion three times in PBS at RT, and they were fixed flat with 30% sucrose in PBS–4% paraformaldehyde, as was described above. Subsequently, tissued cut into ~5 × 5 mm fragments, then were blocked with PBS–3% BSA (Sigma-Aldrich, USA) for 3 h at RT. And to immunostaining, the accessible Ecad, three colonic and one intestinal tissue fragments of each mouse were incubated with a primary polyclonal 1:200 rat anti-E-cadherin (#ab11512; Abcam, USA) in PBS–3% BSA for overnight at 4 °C, then tissues were washed, with PBS, and incubated with 1:300 goat anti-rat IgG secondary antibodies Alexa-Fluor 488 (#A-21470, ThermoFisher, USA) in PBS–3% BSA for 3 h at RT. Subsequently, to perform the

second stain Fn, Vn, or Muc2 in tissues stained for accessible Ecad, all tissues fragments were washed with PBS and then for (i) stain accessible Fn, one colonic tissue fragment of each mice was incubated with 1:200 of rabbit anti-fibronectin (SC9068, Santa Cruz Biotechnologies, USA), or to (ii) stain accessible Vn one colonic tissue fragment of each mice was incubated with 1:200 of rabbit anti-vitronectin (SC15332, Santa Cruz Biotechnologies, USA), and finally, (iii) stain accessible Muc2 in colonic and ileum tissue one tissue fragment of each mice was incubated with 1:200 of rabbit anti-muc2 ab90007 Abcam (#ab90007. Abcam, USA), in PBS–3% BSA for overnight at 4 °C. The next day, tissues were incubated with 1:300 of goat anti-rabbit IgG secondary antibodies Alexa-Fluor 568 (A11036, Invitrogen, USA) for 3 h at RT. Subsequently, tissues were washed and were made permeable by incubation with PBS–0.2% Triton X-100 (Merck, USA) for 1 h at RT. Finally, tissues were incubated with 1:100 phalloidin Alexa-Fluor 647 (A22287 Invitrogen, USA) for 90 min at RT.

The aforementioned immune-stained tissues were subsequently mounted with the luminal side-up. Thus, the colonic crypts and the intestinal villi were identified under light microscopy with ×10 or ×40 magnification and were oriented side-up toward the coverslip. The tissue segment was placed over 5 µL of Dako fluorescent mounting medium (Dako, Denmark) applied onto a glass slide. The tissue covered with 15 µL Dako fluorescent mounting medium and closed with a coverslip. Coverslips were affirmed to the glass slide with vinyl tape to hold the tissue sections in place and were allowed to cure for at least 24 h before imaging.

**Confocal microscopic analysis of ileal and colonic loops**. To acquiring images, two confocal fluorescent microscopes were used; a Leica TCS LSI and a Leica SP8 (Leica, Germany) at the Confocal Microscopy Core Facility of the Universidad Andrés Bello. In the first instance to observe spore internalization in the healthy ileum and colonic mucosa, a Leica TCS LSI was used, with 63× ACS APO oil objective numerical aperture 1.3, and 5× (optical zoom 20×), numerical aperture 0.5. Confocal micrographs were acquired using excitation wavelengths of 405, 488, and 532 nm, and signals were detected with an ultra-high dynamic photomultiplier (PMT) spectral detector (430–750 nm). Emitted fluorescence was split with four dichroic mirrors (QD 405, 488, 561, and 635 nm). Images (1024 × 1024 pixels). To observe the sites with accessible Fn and Vn in the intestinal barrier, and to evaluate the adherence and internalization of the Δ*bclA3* spore mutant to the intestinal barrier or to evaluate spore adherence and internalization in mice treated with RGD and nystatin, confocal images were acquired in Leica SP8 was used with HPL APO CS2 40× oil, numerical aperture 1.30. For signals detection, three (PMT) spectral detectors were used; PMT1 (410–483) DAPI, PMT2 (505–550) Alexa-Fluor 488 and PMT3 (587–726) Alexa-Fluor 555. Emitted fluorescence was split with dichroic mirrors DD488/552. Three-dimensional (3D) reconstructions of intestinal epithelium were performed using ImageJ software (NIH, USA). Villi and crypts were visualized by Hoechst and phalloidin signals. 3D reconstruction movies were performed with software Leica Application Suite X (Leica, Germany).

To quantify cells of the colonic and ileum mucosa with accessible proteins immunodetected, confocal images with a 1-µm Z step size were filtered with Gaussian Blur 3D (sigma *x*: 0.6; *y*: 0.6; *z*: 0.6) and quantifies with cell counting plug-in of ImageJ 1000–1200 cells were counted in an area 84,628 µm$^2$ per mice.

To quantify spore adherence and internalization, confocal images with a 0.7-µm Z step size were analyzed. Adhered spores were considered fluorescent spots in narrow contact with the actin cytoskeleton (visualized with phalloidin). Internalized spores were considered fluorescent spots inside the actin cytoskeleton in the three spatial planes (orthogonal view)[23,55]. The analyzed area for each tissue was 338,512 µm$^2$ per animal.

Then, we evaluated the spore distribution of adhered and internalized in colonic and in the intestinal mucosa. To measure the distribution of adhered spores in the colonic and ileum mucosa, the perpendicular distance from the center of the spore to the epithelium was measured using ImageJ (NHI, USA). In the case of internalized spores, we measured the perpendicular distance from the center of the spore to the mucosa surface or from the closest crypt membrane. For ileum mucosa, we measure the perpendicular distance from the center of the spore to the villus tip or to the villus membrane.

**Visualization of spore internalization in intestinal epithelial cells in vitro by confocal microscopy**. Differentiated Caco-2 cells and T84 cells cultured onto Transwell (Corning, USA) until 1000–2000 Ω. Cells were infected for 5 h with an MOI of 10 with *C. difficile* spores previously stained with Alexa-Fluor 488 Protein Labeling Kit (Molecular Probes, USA), according to the manufacturer's instruction. Cells were washed twice with PBS and were permeabilized with PBS–0.06%-Triton X-100 (Merck, USA) for 10 min at RT, were washed and incubated with 1:150 phalloidin Alexa-Fluor 568 (#A12380 Thermo Fisher, USA) in PBS–1% BSA for 1 h at RT. Then cells were washed, fixed, and visualized in a confocal microscopy Olympus FV1000 of the Confocal Microscopy Core Facility of the Universidad Andrés Bello.

**Sample preparation for transmission electron microscopy and immuno-electron microscopy**. To visualize internalized spores in IECs, six-well plates containing differentiated Caco-2 cells or T84 cells cultured in transwell as was described above were infected for 5 h at 37 °C at an MOI of 20 with *C. difficile*

R20291 spores preincubated 1 h at 37 °C with 100 μL of NHS (Complement, Technology USA) for each well, and then was suspended in the infection volume of 1 mL; FBS final concentration 10% vol vol$^{-1}$ FBS. Unbound spores rinsed off, and cells were scraped, fixed, and processed, as is described below.

To evaluate the binding of Fn and Vn to the surface of *C. difficile* spores, $4 \times 10^7$ *C. difficile* spores were incubated in PBS–0.2% BSA containing 10 μg mL$^{-1}$ of human Fn and Vn for 1 h at 37 °C. The spores were then washed three times ($18,400 \times g$ for 5 min at RT) with PBS. Then the spores were pelleted by one cycle of centrifugation at $18,400 \times g$ for 10 min. Pellets were then resuspended in 200 μl PBS–1% BSA, incubated for 30 min at RT, and then sedimented by centrifugation at $18,400 \times g$ for 10 min at RT. Pellets were resuspended as above and incubated with primary antibody 1:200 rabbit pAb against Fn (SC9068, Santa Cruz Biotechnology, USA) or Vn (SC15332, Santa Cruz Biotechnology, USA) in PBS–BSA 1% for 1 h at RT. The excess of antibody was eliminated by three cycles of centrifugation at $18,400 \times g$ for 5 min at RT and resuspension in PBS–0.1% BSA. Spore suspensions were then incubated for 1 h with 1:20 donkey anti-rabbit IgG antibody coupled to 12-nm gold particles (Abcam ab105295, USA) in PBS–1% BSA for 1 h at RT. And were washed by triple centrifugation at $18,400 \times g$ for 5 min. Subsequently, samples were fixed and processed, as is described below.

To visualize if the collagen-like BclA3 exosporium protein forms the hair-like extension of *C. difficile* spores, ~$2 \times 10^8$ *C. difficile* purified spores of wild-type R20291 (Δ*pyrE/pyrE⁺*), Δ*bclA3*, and Δ*bclA3/bclA3⁺* fixed and processes, as is described here below.

**Sample processing and staining for transmission electron microscopy**. The spore mentioned above or monolayers of infected IECs samples were fixed with freshly prepared with 2.5% glutaraldehyde and 1% paraformaldehyde in 0.1 M cacodylate buffer (pH 7.2) overnight at 4 °C, rinsed in cacodylate buffer, and stained for 30 min with 1% tannic acid. Then samples were serially dehydrated with acetone 30% (with or without 2% uranyl acetate) for 20 min, 50% for 20 min, 75% for 20 min, 90% for 20 min, and twice with 100% for 20 min, embedded in spurs resin at ratio acetone: spurs of 3:1, 1:1, and 1:3 for 40 min each and then resuspended in spurs for 4 h and baked overnight at 65 °C, and prepared for TEM[15]. Thin sections (90 nm) obtained with a microtome were placed on glow discharge carbon-coated grids for negative staining, and double lead stained with 2% uranyl acetate and lead citrate. Grids were analyzed with a Phillips Tecnai 12 Bio Twin electron microscope of the Universidad Católica de Chile.

**R-CDI mouse model**. Antibiotic cocktail (ATB cocktail) an antibiotic cocktail[8] containing 40 mg kg$^{-1}$ kanamycin (Sigma-Aldrich, USA), 3.5 mg kg$^{-1}$ gentamicin (Sigma-Aldrich, USA), 4.2 mg kg$^{-1}$ colistin (Sigma-Aldrich, USA), 21.5 mg kg$^{-1}$ metronidazole (Sigma-Aldrich, USA), and 4.5 mg kg$^{-1}$ vancomycin (Sigma-Aldrich, USA) was administrated via gavage for 3 days (day $-6$ to $-4$ before the infection). Then 1 day before the infection (day $-1$), an intraperitoneal injection of 10 mg kg$^{-1}$ clindamycin (Sigma-Aldrich, USA) was administrated to all mice. The next day all mice were infected via gavage with $1 \times 10^7$ spores R20291. On day 3 post infection, DPBS containing 17,000 UI kg$^{-1}$ of nystatin and 50 mg kg$^{-1}$ vancomycin or vancomycin alone (as control) was orally administered for 5 days to 18 and 23 mice, respectively.

To evaluate the role of the exosporium protein BclA3 in the R-CDI, $n = 40$ mice were treated with antibiotic cocktail followed with clindamycin, and were infected orally with 100 μL of PBS containing $5 \times 10^7$ *C. difficile* spore strain R20291 of wild-type (Δ*pyrE/pyrE⁺*; $n = 10$), Δ*bclA3* ($n = 16$), or Δ*bclA3/bclA3⁺* ($n = 14$) strains. *C. difficile*-infected mice were housed individually in sterile cages with ad libitum access to food and water. All procedures and mouse handling were performed aseptically in a biosafety cabinet to contain spore mediated transmission and cross contamination. Mice were daily monitored for weight loss, aspect, and diarrhea were measured to determine the endpoint of each animal.

The clinical condition (sickness behaviors and fecal samples) of mice was monitored daily with a scoring system during the entire experiment. The presence of diarrhea was classified according to severity as follows: (i) normal stool (score = 1); (ii) color change and/or consistency (score = 2); (iii) presence of wet tail or mucosa (score = 3); and (iv) liquid stools (score = 4). A score higher than 1 was considered as diarrhea[56]. At the end of the assay, animals were sacrificed with a lethal dose of ketamine and xylazine. Cecum content and colonic tissues were collected.

**Quantification of *C. difficile* spores from feces and colon of mice**. To quantify *C. difficile* spores in feces, daily collected fecal samples were stored at $-20$ °C until spore quantification.

Feces were hydrated in 500 μL sterile Milli-Q water overnight at 4 °C and then mixed with 500 μL of absolute ethanol (Merck, USA) for 60 min at RT. Then, serially dilutions of the sample were plated onto selective medium supplemented with 0.1% weight vol$^{-1}$ taurocholate, 16 μg mL$^{-1}$ cefoxitin, 250 μg mL$^{-1}$ L-cycloserine, and 1.5% weight vol$^{-1}$ agar (BD, USA) (TCCFA plates). The plates were incubated anaerobically at 37 °C for 48 h, colonies counted, and results expressed as the log$_{10}$ (CFU g$^{-1}$ of feces)[57]. Colonic tissue collected at the end of the experiment was washed three times with PBS. The tissue *C. difficile* spore load was determined in the proximal colon, medium colon, distal colon, and cecum tissue. Tissues were weighed and

adjusted at a concentration of 100 mg mL$^{-1}$ with a 1:1 mix of PBS:absolute ethanol, then homogenized and incubated for 1 h at RT. The amounts of viable spores were quantified by plating the homogenized tissue onto TCCFA plates, as described previously[18]. The plates were incubated anaerobically at for 48 h at 37 °C. Finally, the colony count was expressed as the log$_{10}$ (CFU g$^{-1}$ of the tissue).

**Cecum content cytotoxicity assay in Vero cells of infected mice during R-CDI**. A 96-well flat-bottom microtiter plates were seeded with Vero cells at a density of $10^5$ cells well$^{-1}$. Mice cecum contents were suspended in PBS at a ratio of 1:10 (100 mg mL$^{-1}$ of cecum content), vortexed, and centrifuged at $18,400 \times g$ for 5 min, the supernatant was sterilized with a 0.22 μm filter and serially diluted in DMEM supplemented with 10% vol vol$^{-1}$ FBS and 100 U mL$^{-1}$ penicillin, and 100 μg mL$^{-1}$ streptomycin; then 100 μL of each dilution was added to wells containing Vero cells. Plates were screened for cell rounding 16 h after incubation at 37 °C. The cytotoxic titer was defined as the reciprocal of the highest dilution that produced rounding in at least 80% of Vero cells per gram of luminal samples under ×20 magnification.

**Statistical analysis**. Prism 7 (GraphPad Software, Inc.) was used for statistical analysis. Student's *t* test and the nonparametric test was used for pairwise comparison. Significance between groups was done by Mann–Whitney unpaired *t* test. Comparative study between groups for in vitro experiments was analyzed by analysis of variance with post hoc Student's *t* tests with Bonferroni corrections for multiple comparisons, as appropriate. A *P* value of ≤0.05 was accepted as the level of statistical significance. Differences in the percentages of mice with normal stools, as well as percentages of mice with CDI, were determined by Gehan–Breslow–Wilcoxon test.

**Reporting summary**. Further information on research design is available in the Nature Research Reporting Summary linked to this article.

## Data availability
The data that support the findings of this study are available in this article and Supplementary Materials, or from the corresponding author upon request. Source data are provided with this paper.

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

## Acknowledgements

This work was funded by FONDECYT Regular 1191601, FONDECYT Regular 1151025, and Millennium Science Initiative Program–NCN17_093. P.C.-C. had been supported by ANID doctoral fellowship 21161395 (Chile) and J.O.-A. by OAICE-91-2018 of Universidad de Costa Rica. The authors acknowledge Rosario Hernandez-Armengol for technical assistance and Miriam Barros for useful discussion on image processing at the Confocal Microscopy Core Facility of the Universidad Andrés Bello. We certify that funding sources had no implication in the study design, data collection, analysis, and interpretation of data.

## Author contributions

Study concept and design: P.C.-C., M.P.-G., and D.P.-S. Acquisition of data: P.C.-C., P.M.-U., R.R.-R., G.C.-A., J.O.-A., M.M.-L., C.B.-S., M.P.-G., and D.P.-S. Analysis and interpretation of the data: P.C.-C., P.M.-U., M.P.-G., G.C.-A., and D.P.-S. Drafting of the manuscript: D.P.-S. Critical revision of the manuscript for important intellectual content: P.C.-C. and D.P.-S. Statistical analysis: P.C.-C., P.M.-U., G.C.-A., and D.P.-S. Obtained funding: D.P.-S. Technical or material support: D.P.-S., N.M., and S.K. Study supervision: D.P.-S.

## Competing interests

D.P.-S. and P.C.-C. are inventors on the PCT WO2020035720 (A1) patent relating to a method and pharmacological composition for the prevention of recurrent infections caused by *Clostridioides difficile*, submitted by Universidad Andrés Bello. The other authors declare no competing interests.

**Additional information**

