## [Peer Review File · Nature Communications]

Reviewers' comments:

Reviewer #1 (Remarks to the Author):

Authors reported that *C. diff* spores can gain entry into intestinal epithelial cells (IECs) via fibronectin- $\alpha 5 \beta 1$ and vitronectin- $\alpha v \beta 1$ specific pathways. They further claimed that spore-surface exosporium BclA3 protein is essential for both spore-entry pathways into IECs. Their data indicates that the intracellular *C. diff* spores act as a reservoir, favoring spore-persistence and recurrence of the disease. The data provided are convincing and supportive for the claims. These findings are novel in *C. diff* pathogenesis, no such reports previously, to my knowledge.

However, the manuscript did not address /discuss several questions below to help the audience understand the depth of the claims:

- 1) No in vivo *C. diff* infection data confirming the claims. i.e., spore entry into IECs in vivo infection model.
- 2) No data shows that the cells into which *C. diff* spores enter are IECs? They might be immune cells? if it is, not surprising that immune cells can take up *C. diff* spores
- 3) How serum can facilitate spore entry into IECs? The data indicates that Fn/Vn in sera do the job? Why not use human sera? Human sera is available in the gut to help *C. diff* spore entry into gut cells?
- 3) If spores enter into IECs, will be resistant to germination due to the in availability to germinants (bile acids), why bother? Spores will be released by the IECs to start recurrent *C. diff* infection? Why and how released? any hypothesis?
-) It would be more interesting and significant to quantitate spores remained in the loop/gut and spores entry into cells. The ratio would tell us which source of spores are the major cause of recurrent CDI

Reviewer #2 (Remarks to the Author):

The paper presented by Castro-Córdova et al., "Intracellular spores of an enteric pathogen are a reservoir for recurrent infections," seeks to present data that *Clostridioides difficile* spores become internalized by host and pathogen factors. More specifically, entry into epithelial cells is dependent on the host glycoproteins, fibronectin- $\alpha 5 \beta 1$ and vitronectin- $\alpha v \beta 1$ while the spore collagen-binding protein, BclA3, is involved in epithelial cell entry/internalization. Authors extensively show using confocal microscopy that *C. difficile* spores are internalized into epithelial cells both in vitro and in vivo. The authors use pharmacological and antibody mediated blockade to identify the host factors and genetic mutation/complementation strategies to demonstrate the involvement of BclA3. Importantly, the authors use a recurrence model of *C. difficile* infection to illustrate the impact of spore entry/internalization on relapsing disease.

This reviewer feels that the scientific impact of the work, specifically that the intestinal epithelium can serve as a reservoir for *C. difficile* spores on relapsing *C. difficile* disease would be greatly enhanced with additional in vivo work. Use of additional rounds of vancomycin treatment to induce relapsing disease would show persistence as a reservoir (as in reference 9). Also, if spores are internalized, wouldn't you expect to see spores within epithelial cells after vancomycin treatment, when they are undetectable in stool by traditional plate counts? And if they do persist intracellularly, how long does this occur for? The authors will also need to consider the impact of environmental transmission (through cage/fur/etc.). An additional challenge that is worth consideration is the impact of toxin production since the toxins alone are responsible for much of the pathophysiology in vivo. The authors make mention of toxin activity potentially contributing to damage and thereby persistence. One consideration to focus on the spore aspect of the work would be to use a nontoxicogenic strain (which is not without potential caveats either).

With regards to the animal experiments, the weight change noted over the first three days (Ex Figs 17 & 22) is less than would be expected for this model – it is possible that because this is a breeding colony, the microbiome impacts the model.

Fig 5; having panels F-I right below the NYS/VAN experiment is confusing – consider reorienting the panels so the experimental timeline for the knock-out experiment becomes panel F. Panel M is misleading to readers since the data in Ex Fig 22, D-F show no significant differences. Is there significance to the medium colon as opposed to portions proximal and distal to this site for adherence?

Extended data Fig 5; switch the orientation of the bars on the villus tip/surface (make vertical) and membrane/crypt (make horizontal) to match your illustrations for reader clarity.

Extended data Fig 7; the use of HEp-2 cells does not add additional information to the figure. In the reporting summary under cell lines, HEp-2 cells are on the list of commonly misidentified cell lines, so this just potentially detracts. In addition, the legends in B & C may be incorrect – should this be FBS rather than SFB?

Extended data Fig 22; please include the numbers of animals in the figure legend. In the methods section (related to this experiment; lines 303-06), the authors state that the experiment contained 50 mice, however, there are only 40 in the numbers listed. Please reconcile this.

Methods; line 105 – please define what portion of the ileum was sampled.

Methods: lines 118-121. The fluorophores used for your antibodies here does not match your figures (i.e., it appears that the colors for spores and actin are switched in confocal micrographs). If you have digitally reassigned the colors, please state that here.

Methods: lines 300-302; no information on numbers of animals used in the experiment is given – there are numbers in the figure legends, but the numbers in Fig 5 and extended Fig 17 do not match. If there is lethality/euthanasia as deemed by humane endpoints, it is worth mentioning for this experiment and the knock out experiment either in the main body of the figure or as supplemental information.

Point by Point Response to reviewer.

Reviewers' comments:

Reviewer #1 (Remarks to the Author):

Authors reported that *C.diff* spores can gain entry into intestinal epithelial cells (IECs) via fibronectin- α 5 β 1 and vitronectin- α v β 1 specific pathways. They further claimed that spore-surface exosporium BclA3 protein is essential for both spore-entry pathways into IECs. Their data indicates that the intracellular *C diff* spores act as a reservoir, favoring spore-persistence and recurrence of the disease. The data provided are convincing and supportive for the claims. These findings are novel in *C.diff* pathogenesis, no such reports previously, to my knowledge.

However, the manuscript did not address /discuss several questions below to help the audience understand the depth of the claims:

1) No in vivo *C diff* infection data confirming the claims. i.e., spore entry into IECs in vivo infection model.

Author response: we acknowledge the reviewer's observation and agree that in the current experiments we are not showing internalized *C. difficile* spores derived from the infection. We have eliminated this claim and provide associations that lead to support the notion of *C. difficile* spore-entry into IECs *in vivo*.

The amount of *C. difficile* spores that might reside in the intestinal tissue during vancomycin treatment is so low, that we are not able to detect it by plate counts. We are currently developing a genetic device that will allow *in vivo* tagging and detection of low levels *C. difficile* spores in intestinal tissue. We are also currently developing protocols for two-photon microscopy to analyze the entire intestinal tract of *C. difficile* infected mice.

However, we incorporated an intestinal loop experiment where we demonstrate reduced *C. difficile* spore-entry into the intestinal mucosa in the presence of the cholesterol-sequestering agent, nystatin (Fig. 7d-j). This correlates with *in vitro* results (Fig. 7A) showing that nystatin reduces spore-entry into IECs in a dose dependent manner, and further supports the notion that sequestering cholesterol rafts, leads to reduced spore-entry and increased spore-germination in the presence of taurocholate (Fig. 7c). When these results are coupled with *in vitro* observations (Fig. 7c) that demonstrate that blocking spore-entry in the presence of germinants leads to complete spore germination, we can further speculate that blocking spore-entry during infection leads to extracellular spores to germinate and become susceptible to vancomycin germination. Please see lines 339-392. Also Figs 7a-j and Extended data Fig18a-f and in the discussion section in lines 477-487.

We have also incorporated an intestinal loop experiment where we demonstrate that the presence of the RGD peptide reduces spore-entry, but not adherence, to the intestinal mucosa, suggesting that spore-entry *in vivo* could be Fn- and Vn-integrin mediated. The depth of these results is also explained in the discussion section. Specifically, in lines 371-381; 443-449. Figs 7d-j and Extended data Fig 8d-f.

In addition, we have substantially modified the manuscript to help the audience understand the depth of these claims. In particular, we have lowering the claim that "intracellular *C diff* spores act as a reservoir, favoring spore-persistence and recurrence of the disease", emphasizing those claims that are supported by the *in vitro* and *in vivo* data.

2) No data shows that the cells into which *Cdiff* spores enter are IECs? They might be immune cells? if it is, not surprising that immune cells can take up *cdiff* spores

Author response: We acknowledge this observation and agree that we need to clarify to the audience that, at this stage, we did not evaluate the cell type(s) involved in spore-entry into the intestinal epithelia. However, in an effort to start identifying potential sites in the intestinal barrier that might facilitate *C. difficile* spore-entry, we have contextualized our new data from immunofluorescence of healthy mice intestine showing that most cells that have accessible fibronectin and vitronectin have accessible E-cadherin (Fig. 3e-n) (please see lines 185-213) (see answers below). We have also discussed this extensively the potential cell-types that can have accessible molecules such as fibronectin,

vitronectin and/or integrins that may contribute to spore-entry into the intestinal mucosa. Please see lines: 430-443.

3) How serum can facilitate spore entry into IECs? The data indicates that Fn/Vn in sera do the job? Why not use human sera? Human sera is available in the gut to help *C. diff* spore entry into gut cells?

Author response: We acknowledged that despite the fact that experiments answering the first three questions were present, we did not adequately address the interpretation of the data to response to these three questions. We have made major modifications in the text (See lines 165-183) to address these questions.

To address the last of these four questions (Human sera is available in the gut to help *C. difficile* spore entry into gut cells?), we have incorporated results from staining of gut mucosa where we first identify the population of cells that have accessible Fn and Vn, and within these acc Fn+ and acc Vn+ cells we quantify the percentage of cells with accessible E-cadherin, which is indicative of sites in the intestinal epithelial that is undergoing cell extrusion or that correspond to epithelial folds where the membrane forces lead to a disruption of the adherent junctions and partial loss of the polarization of the cells. We have further identified that a fraction of Acc-Ecadherin positive cells is identified as goblet cells, suggesting that these are also cell-type candidates that might contribute to *C. difficile* spore-entry into the intestinal mucosa. These results demonstrate potential sites where *C. difficile* spores could interact with Fn and Vn to gain spore-entry into IECs. Please see Fig. 3E-N, and lines 185 - 213.

4) If spores enter into IECs, will be resistant to germination due to the in availability to germinants (bile acids), why bother? Spores will be released by the IECs to start recurrent *C. diff* infection? Why and how released? any hypothesis?

Author response: The experiments that answers this question were already in place in the previous version. In the previous discussion section, we also indicated our hypothesis: that *C. difficile* spores are likely to be released to the environment due to intestinal epithelial turnover, which in humans is estimated to occur every 5 days.

In our previous version we showed that internalized *C. difficile* spores are resistant to germination. This data was previously located in extended figures. In the re-submitted version, it has been moved to figure 7C. These results demonstrate that, in the presence of germinant, intracellular spores do not germinate *in vitro*, which would suggest that during the infection, intracellular spores would not be inactivated by vancomycin treatment. Thus, once the epithelial cells that harbor intracellular *C. difficile* spores is extruded due to epithelial turnover, they might contribute to the recurrence of the infection.

We have enhanced our writing to address these aspects for better clarity to the audience.

5) It would be more interesting and significant to quantitate spores remained in the loop/gut and spores entry into cells. The ratio would tell us which source of spores are the major cause of recurrent CDI

Author response: We agree with the reviewer and therefore, we have changed the panels so that the results from the ileal/gut model show percentage of total adhered cells. Interestingly, we now observe that while *C. difficile* spore-entry into IECs *in vivo* is still RGD and nystatin-specific; by contrast, absence of BclA3 leads to a significant reduction in adherence to the intestinal mucosa, but not for spore-entry in the ileal loop mouse model. These last results are in disagreement with our *in vitro* data and raises further outstanding questions on whether additional spore-ligands are contributing to spore-entry into the intestinal mucosa.

Please see figs 6d-g and 7g-j and lines 297-308 and 371-381.

Reviewer #2 (Remarks to the Author):

The paper presented by Castro-Córdova et al., "Intracellular spores of an enteric pathogen are a reservoir for recurrent infections," seeks to present data that *Clostridioides difficile* spores become internalized by host and pathogen factors. More specifically, entry into epithelial cells is dependent on the host glycoproteins, fibronectin- $\alpha 5\beta 1$ and vitronectin- $\alpha v\beta 1$ while the spore collagen-binding protein, BclA3, is involved in epithelial cell entry/internalization. Authors extensively show using confocal microscopy that *C. difficile* spores are internalized into epithelial cells both in vitro and in vivo. The authors use pharmacological and antibody mediated blockade to identify the host factors and genetic mutation/complementation strategies to demonstrate the involvement of BclA3. Importantly, the authors use a recurrence model of *C. difficile* infection to illustrate the impact of spore entry/internalization on relapsing disease.

This reviewer feels that the scientific impact of the work, specifically that the intestinal epithelium can serve as a reservoir for *C. difficile* spores on relapsing *C. difficile* disease would be greatly enhanced with additional in vivo work. Use of additional rounds of vancomycin treatment to induce relapsing disease would show persistence as a reservoir (as in reference 9). Also, if spores are internalized, wouldn't you expect to see spores within epithelial cells after vancomycin treatment, when they are undetectable in stool by traditional plate counts? And if they do persist intracellularly, how long does this occur for? The authors will also need to consider the impact of environmental transmission (through cage/fur/etc.). An additional challenge that is worth consideration is the impact of toxin production since the toxins alone are responsible for much of the pathophysiology in vivo. The authors make mention of toxin activity potentially contributing to damage and thereby persistence. One consideration to focus on the spore aspect of the work would be to use a nontoxigenic strain (which is not without potential caveats either).

Author response: We appreciate these questions, as they naturally arise from our presented data and will lead to further work that require the development of new in vivo tools (animal models) to address these questions. Having multiple rounds of relapsing *C. difficile* disease, although would greatly enhance the results, as stated by the reviewer is very challenging due to environmental transmission (through cage/fur/etc). This will definitively be a useful experimental platform to assess the contribution of virulence factors to recurrence of CDI. An intriguing question is, if spores are undetectable by traditional plate counts, will they be in sufficient numbers to be detectable by imaging techniques?. To address this question, we are developing new tools for *in vivo* imaging and in vivo tagging. In this context, two-photon confocal microscopy is key to assess this important question raised by the reviewer; we are currently lacking this equipment. This is followed by the question of how long do *C. difficile* spores persist intracellularly?; we provided an answer to reviewer one (see above), and we clearly need to develop a tagging system to monitor intracellular spores during the infection to assess the intracellular timing prior to their contribution to recurrence of the infection; these are major technical and experimental challenges. As further questioned by the reviewer, a recent article has been published demonstrating that massive epithelial damage contributes to the recurrence of the infection (Mileto et al 2020, PNAS April 7). Thus, to address the question adequately, we would require to titer toxin production in vivo, to correlate damage versus persistence in the various mutant strains. In summary, these outstanding questions scientifically sound, and will drive our future work to fully understand the mechanism of *C. difficile* spore-mucosal interaction that contributes to the persistence of the disease.

To clarify the depth of our claims to the audience, we have made significant modifications in the text. We have also included several new experiments that will provide in vivo support to our in vitro claims.

With regards to the animal experiments, the weight change noted over the first three days (Ex Figs 17 & 22) is less than would be expected for this model – it is possible that because this is a breeding colony, the microbiome impacts the model.

Author response: Our results are consistent with those previously reported for the animal model of the disease using strain R20291. It is well established that infection of mice with strain R20291 causes a subclinical colitis, which leads to reduced weight loss during the infection in comparison to other hypervirulent strains.

Our results are consistent with those that follow:

1. Castro-Cordova P, Diaz-Yanez F, Munoz-Mirallas J, Gil F, Paredes-Sabja D. Effect of antibiotic to induce *Clostridioides difficile*-susceptibility and infectious strain in a mouse model of *Clostridioides difficile* infection and recurrence. *Anaerobe*. 2020; 62:102149. PMID:31940467
2. Winston JA, Thanissery R, Montgomery SA, Theriot CM. Cefoperazone-treated Mouse Model of Clinically-relevant *Clostridium difficile* Strain R20291. *J Vis Exp*. 2016;(118).
3. Pizarro-Guajardo, M, Fernando-Díaz, F., Álvarez-Lobos, M, Paredes-Ssabja, D. Characterization of Chicken IgY specific to *Clostridium difficile* R20291 spores and the effect of oral administration in mouse models of initiation and recurrent disease. *Front Cell Infect Microbiol*. 2017. 7:365. PMID: 28856119
4. Muñoz-Mirallas J, Trindade BC, Castro-Córdova P, Bergin IL, Kirk LA, Gil F, Aronoff DM, Paredes-Sabja D. Indomethacin increases severity of *Clostridium difficile* infection in mouse model. *Future Microbiol*. 2018 Sep;13(11):1271-1281. doi: 10.2217/fmb-2017-0311. Epub 2018 Sep 21. PMID: 30238771

Fig 5; having panels F-I right below the NYS/VAN experiment is confusing – consider reorienting the panels so the experimental timeline for the knock-out experiment becomes panel F. Panel M is misleading to readers since the data in Ex Fig 22, D-F show no significant differences. Is there significance to the medium colon as opposed to portions proximal and distal to this site for adherence?

Author response: We have made changes as requested by the reviewer. Also, we have re-organized these figures under three different main groups: Figure. 5 is showing the impact of BclA3 in spore-entry in vitro; Figure 6 is showing the role of BclA3 during in vivo spore-adherence and spore-entry, as well as in the animal model of recurrence of the disease; and finally, Figure 7, shows that inhibition of spore-entry leads to increased germination, that nystatin and RGD block in vivo spore-entry and that nystatin reduces the recurrence in mice when given together with vancomycin.

We have also indicated the differences in the medium colon of mice in the discussion section. Basically, the distal and proximal colon have mucosal folds, while the medium colon is rather flat in its mucosal topology. Please see lines: 471-473.

Extended data Fig 5; switch the orientation of the bars on the villus tip/surface (make vertical) and membrane/crypt (make horizontal) to match your illustrations for reader clarity.

Author response: Appreciated, we have changed as requested.

Extended data Fig 7; the use of HEp-2 cells does not add additional information to the figure. In the reporting summary under cell lines, HEp-2 cells are on the list of commonly misidentified cell lines, so this just potentially detracts. In addition, the legends in B & C may be incorrect – should this be FBS rather than SFB?

Author response: We agree, we have removed these figures to avoid distractions to the audience as suggested.

Extended data Fig 22; please include the numbers of animals in the figure legend. In the methods section (related to this experiment; lines 303-06), the authors state that the experiment contained 50 mice, however, there are only 40 in the numbers listed. Please reconcile this.

Author response: We appreciate this observation; we have included the number of animals in the figure legend, which should now match the method section.

Methods; line 105 – please define what portion of the ileum was sampled.

Author response: We appreciate this observation; the information was added. Please see line:750-751.

Methods: lines 118-121. The fluorophores used for your antibodies here does not match your figures (i.e., it appears that the colors for spores and actin are switched in confocal micrographs). If you have digitally reassigned the colors, please state that here.

Author response: We apologize for this incongruency. We have made corrections. In methods we describe the fluorophores (See lines 775 - 810), and in figure legend we have added the colors. Please see Legends of: Fig 1, Fig 2, Fig 3 Fig 6, Fig 7, Ext. data fig 1, Ext. data fig 2, Ext. data fig3, Ext. data fig 4, Ext. data fig 6, Ext. data fig 10, Ext. data fig 17, Ext. data fig 18.

Methods: lines 300-302; no information on numbers of animals used in the experiment is given – there

are numbers in the figure legends, but the numbers in Fig 5 and extended Fig 17 do not match. If there is lethality/euthanasia as deemed by humane endpoints, it is worth mentioning for this experiment and the knock out experiment either in the main body of the figure or as supplemental information.

Author response: Thank you. We have corrected the numbers of animals and have provided details in the methods text (Please see lines: 917-918 and 921-922) as well as in figure legend of extended figure 18 (see line 1475).

REVIEWERS' COMMENTS

Reviewer #1 (Remarks to the Author):

Authors have adequately addressed this reviewer's questions in the revised manuscript.

Minor issues.

1) It seems that authors did not discuss about the different adhesion effects of BclA3 in in vitro epithelial cells and loop models.

2) The figures are too big to follow, and the legends are too long to follow.

Reviewer #2 (Remarks to the Author):

The authors have added significant experimental information that contribute to and strengthen the study. Moreover, additions and modifications of the text convey the strength and future translational potential for the outlined study. The images offer compelling information to the validity of the mechanism for spore internalization presented here. The author appreciates the amount of work that has been done, as the procedures and techniques used are not trivial.

There are a few minor edits to clarify text:

Line 44: consider changing intestinal mucosa to intestinal milieu

Line 100: change to luminally

Line 101: The sentence is incomplete

Line 305-307: Is there the potential for some uncharacterized host factor that may also contribute?

If plausible, please add inclusive wording as appropriate.

Line 760: If there were any post-surgical analgesic used and is not covered in reference 18, please list.

Line 984: change to Clostridioides

Line 1183: Does acc mean accessible? If so, please spell out for the first time use and elsewhere as appropriate.

Signed: Glynis L. Kolling, Ph.D.